# Repressor element 1-silencing transcription factor deficiency yields profound hearing loss through $K_v7.4$ channel upsurge in auditory neurons and hair cells

**Haiwei Zhang[1,2,3], Hongchen Li[1,2,3], Mingshun Lu[1,2,3], Shengnan Wang[1,2,3], Xueya Ma[1,2,3], Fei Wang[1,2,3], Jiaxi Liu[1,2,3], Xinyu Li[1,2,3], Haichao Yang[1,2,3], Fan Zhang[2,3], Haitao Shen[4], Noel J Buckley[5,6], Nikita Gamper[1,2,7], Ebenezer N Yamoah[8]\*, Ping Lv[1,2,3,9]\***

[1]Department of Pharmacology, The Key Laboratory of New Drug Pharmacology and Toxicology, Hebei Medical University, Shijiazhuang, China; [2]Center for Innovative Drug Research and Evaluation, Institute of Medical Science and Health, Hebei Medical University, Shijiazhuang, China; [3]The Key Laboratory of Neural and Vascular Biology, Ministry of Education, Hebei Medical University, Shijiazhuang, China; [4]Lab of Pathology, Hebei Medical University, Shijiazhuang, China; [5]Department of Psychiatry, University of Oxford, Oxford, United Kingdom; [6]Kavli Institute for Nanoscience Discovery, University of Oxford, Oxford, United Kingdom; [7]Faculty of Biological Sciences, University of Leeds, Leeds, United Kingdom; [8]Department of Physiology and Cell Biology, School of Medicine, University of Nevada, Reno, United States; [9]The Hebei Collaboration Innovation Center for Mechanism, Diagnosis and Treatment of Neurological and Psychiatric Disease, Hebei Medical University, Shijiazhuang, China

**\*For correspondence:**
enyamoah@gmail.com (ENY);
lping77@hotmail.com (PL)

**Competing interest:** The authors declare that no competing interests exist.

**Abstract** Repressor element 1-silencing transcription factor (REST) is a transcriptional repressor that recognizes neuron-restrictive silencer elements in the mammalian genomes in a tissue- and cell-specific manner. The identity of REST target genes and molecular details of how REST regulates them are emerging. We performed conditional null deletion of *Rest* (cKO), mainly restricted to murine hair cells (HCs) and auditory neurons (aka spiral ganglion neurons [SGNs]). Null inactivation of full-length REST did not affect the development of normal HCs and SGNs but manifested as progressive hearing loss in adult mice. We found that the inactivation of REST resulted in an increased abundance of $K_v7.4$ channels at the transcript, protein, and functional levels. Specifically, we found that SGNs and HCs from *Rest* cKO mice displayed increased $K_v7.4$ expression and augmented $K_v7$ currents; SGN's excitability was also significantly reduced. Administration of a compound with $K_v7.4$ channel activator activity, fasudil, recapitulated progressive hearing loss in mice. In contrast, inhibition of the $K_v7$ channels by XE991 rescued the auditory phenotype of *Rest* cKO mice. Previous studies identified some loss-of-function mutations within the $K_v7.4$-coding gene, *Kcnq4*, as a causative factor for progressive hearing loss in mice and humans. Thus, the findings reveal that a critical homeostatic $K_v7.4$ channel level is required for proper auditory functions.

## Editor's evaluation

Genetic forms of deafness are a major health challenge. This study deciphers the cochlear roles of repressor element 1–silencing transcription factor (REST), a gene involved in the DFNA27 dominant form of deafness, using the mouse as a model system. This study provides evidence for a pathophysiological mechanism of deafness and shows how genes involved in different forms of deafness may interact together. The article will be interesting to readers who work in the field of hearing research, REST regulation, or $K_v7.4$ regulation.

## Introduction

The repressor element 1-silencing transcription factor (REST), also known as neuronal restriction-silencing factor (NRSF), is a transcriptional repressor that binds a specific 21 bp consensus sequence named repressor element 1 (RE-1) (*Chong et al., 1995*; *Schoenherr and Anderson, 1995*). REST recruits multiple chromatin remodeling factors through its DNA-binding domain to form a repressor complex, ultimately repressing the transcription of target genes (*Ooi and Wood, 2007*). REST is expressed in embryonic stem cells, neural stem cells, and non-neural cells during embryogenesis. It maintains embryonic stem cells' pluripotency and promotes stem cell differentiation and neuronal development, which are essential for neuronal diversity, plasticity, and survival (*Chen et al., 1998*; *Sun et al., 2005*). In non-neuronal cells, REST maintains a non-neuronal cell gene expression pattern by suppressing the expression of neural-related genes through histone deacetylation, chromatin remodeling, and methylation (*Chen et al., 1998*). Recent studies suggest that the role of REST expands beyond neuronal development. There are nearly 2000 genes containing predicted REST binding sites (*Bruce et al., 2004*; *Johnson et al., 2007*; *Seki et al., 2014*), and up to 90% of these REST target genes are tissue- and cell-type-specific (*Bruce et al., 2009*; *Hohl and Thiel, 2005*). As a result, REST is involved in numerous physiological and pathological processes. For example, REST was suggested to have a neuroprotective role in Alzheimer's disease (*Lu et al., 2014*). Upregulation of REST in peripheral somatosensory neurons was shown to contribute to the development of neuropathic pain (*Rose et al., 2011*; *Zhang et al., 2018*; *Zhang et al., 2019*). In addition, longevity in humans was associated with REST upregulation and repression of neuronal excitation (*Zullo et al., 2019*).

Cochlear hair cells (HCs) and spiral ganglion neurons (SGNs) play essential roles in transmitting auditory signals. Inner hair cells (IHCs) convert mechanical stimuli into electrochemical signals transmitted to and conducted along auditory nerve fibers (*Ó Maoiléidigh and Ricci, 2019*; *Fettiplace, 2017*; *Yu and Goodrich, 2014*). In contrast, outer hair cells (OHCs) act as mechanical amplifiers that enhance the sensitivity to weak sounds in the cochlea (*Dallos et al., 2008*; *Liberman et al., 2002*). SGNs are at the bottleneck between HCs and the brain, preserving sound information's amplitude, frequency, and temporal features (*Meyer and Moser, 2010*; *Taberner and Liberman, 2005*). Since HCs and SGNs in the mature mammalian cochlea do not regenerate, their damage or dysfunction leads to permanent sensorineural hearing loss (*Fujioka et al., 2015*).

A recent study reported that the exon 4-containing REST splice variant (*Rest*[4]) is expressed in cochlear HCs and that its deletion results in abnormally high REST expression and is associated with DFNA27 hearing loss (*Nakano et al., 2018*). Yet, the role of the full-length REST in hearing is still unknown. We generated a mouse model with REST conditionally knocked out in the cochlea. We show that REST is expressed in cochlear HCs and SGNs of adult mice and that the conditional deletion of REST in these cells results in progressive hearing loss. No detectable loss of cochlear HCs and SGNs was found at P1–P14 during development, but significant degeneration of OHCs and SGNs was detected at 3 months. Examination of the function of HCs and SGNs showed that $K_v7.4$ channel expression was upregulated, reducing SGNs' excitability. Consistently, administration of the compound with $K_v7.4$ channel activator activity, fasudil, caused hearing loss in wild-type mice, while inhibition of the $K_v7$ channels by XE991 rescued the hearing phenotype of *Rest* cKO mice. In summary, our results demonstrate that REST is essential for hearing and its deficiency causes upregulation of $K_v7.4$ channels contributing to the dysfunction of SGNs and HCs and deafness in mice.

## Results

### REST is expressed in SGNs and HCs and is required for hearing

To examine the role of REST in hearing, we first examined the expression of REST in the cochlea of 1-month-old wild-type (WT) mice. We found that REST was abundantly expressed in cochlear SGNs, IHCs, and OHCs but absent in *Rest* cKO mice as determined by immunofluorescence staining (*Figure 1A*) and single-cell PCR (*Figure 1—figure supplement 1A*). *Rest* and its splice form *Rest⁴* were detected by single-cell RT-PCR in OHCs on P13, IHCs on P20, and SGNs on P40 in WT mice (*Figure 1B*). We then investigated whether specific deletion of *Rest* in the cochlea affected mouse hearing. Mice with homozygous intronic LoxP sites flanking exon 2 of *Rest* (*Rest^flox/flox*) (*Soldati et al., 2012*) were crossed with *Atoh1-Cre* mice (*Matei et al., 2005*; *Yang et al., 2010*). The Cre was expressed in HCs and SGNs and some supporting cells in the inner ear (*Matei et al., 2005*; *Yang et al., 2010*), resulting in the specific knocking out of *Rest* in these cells (*Figure 1C*). The genotypes of the pups were identified using PCR analysis (*Figure 1D*).

The auditory function of 1–2-month-old WT and *Rest* cKO animals was evaluated by measuring auditory brainstem responses (ABR) to click and pure-tone stimuli. Both heterozygous (*Rest* +/cKO) and homozygous *Rest* knockout (*Rest* cKO) mice exhibited a significant increase in the ABR threshold (as compared to WT), indicating hearing impairment (*Figure 1E–G*). There was no statistical difference in hearing impairment between *Rest* +/cKO and *Rest* cKO mice. Therefore, we used *Rest* cKO mice in subsequent experiments.

We evaluated the hearing function of 1- and 3-month-old *Rest* cKO mice. Compared to age-matched WT mice, both 1- and 3-month-old *Rest* cKO mice displayed a significantly elevated ABR threshold across all frequencies (*Figure 1H–J*). Furthermore, in 3-month-old *Rest* cKO mice, there were no detectable ABRs at 24 kHz, 28 kHz, and 32 kHz, suggesting complete hearing loss in the high-frequency range (*Figure 1J*). These results indicate that *Rest* cKO mice undergo progressive hearing loss. Similar results were observed in *Rest* +/cKO mice (*Figure 1—figure supplement 1B and C*). ABRs typically consist of five positive waves: wave I represents activity from the auditory nerve, while waves II–V represent neural transmission within the central auditory system. ABR latencies and amplitudes of waves I and II were measured with a click stimulus and a 16 kHz pure-tone stimulus in WT and *Rest* cKO mice to assess hearing sensitivity and functional integrity of the auditory nerve. Compared to WT mice, 1-month-old *Rest* cKO mice had stimulus level-dependent decrease in amplitude and increase in latency of wave I. This effect was even more pronounced in 3-month-old mice, indicating a gradual decline in the hearing status of *Rest* cKO mice (*Figure 1—figure supplement 2A–D*). Similarly, we observed a significant difference in the amplitude and latency of wave II between WT and *Rest* cKO mice at 1 and 3 months of age (*Figure 1—figure supplement 2E–H*). These results indicate a deficit in auditory neural processing in *Rest* cKO mice; this decline in hearing status ensues from early sound processing at the levels of HCs and SGNs.

### Degeneration of SGNs and HCs is observed in 3-month-old but not in 1-month-old *Rest* cKO mice

To investigate the cause of hearing loss induced by REST deficiency, we examined the morphology of SGNs and HCs in *Rest* cKO mice. Immunofluorescence and HE staining revealed no apparent morphological abnormalities or SGN and HC loss in the apical, middle, and basal turns in P1–14 and 1-month-old *Rest* cKO cochleas (*Figure 2B and F*, *Figure 2—figure supplement 1*). However, SGN and HC degeneration was observed in the cochlea of 3-month-old mice (*Figure 2C–E and G–L*). These data indicate that SGNs and HCs loss might be responsible for hearing loss in 3-month-old *Rest* cKO mice but perhaps not in 1-month-old mice.

### Excitability of SGNs is decreased in *Rest* cKO mice

Given the lack of apparent indications of morphological alterations in SGNs and HCs in 1-month-old *Rest* cKO mice, we explored whether dysfunction of SGNs or HCs resulted in the early onset of hearing impairment in these mice. We first examined the excitability of SGNs in *Rest* cKO and WT mice. The whole-cell patch-clamp technique recorded action potentials (APs) in SGNs. A 0.4 nA current injection evoked a single spike in the apical SGNs of the cochlea in WT mice (*Figure 3A*). In contrast, 27.8%

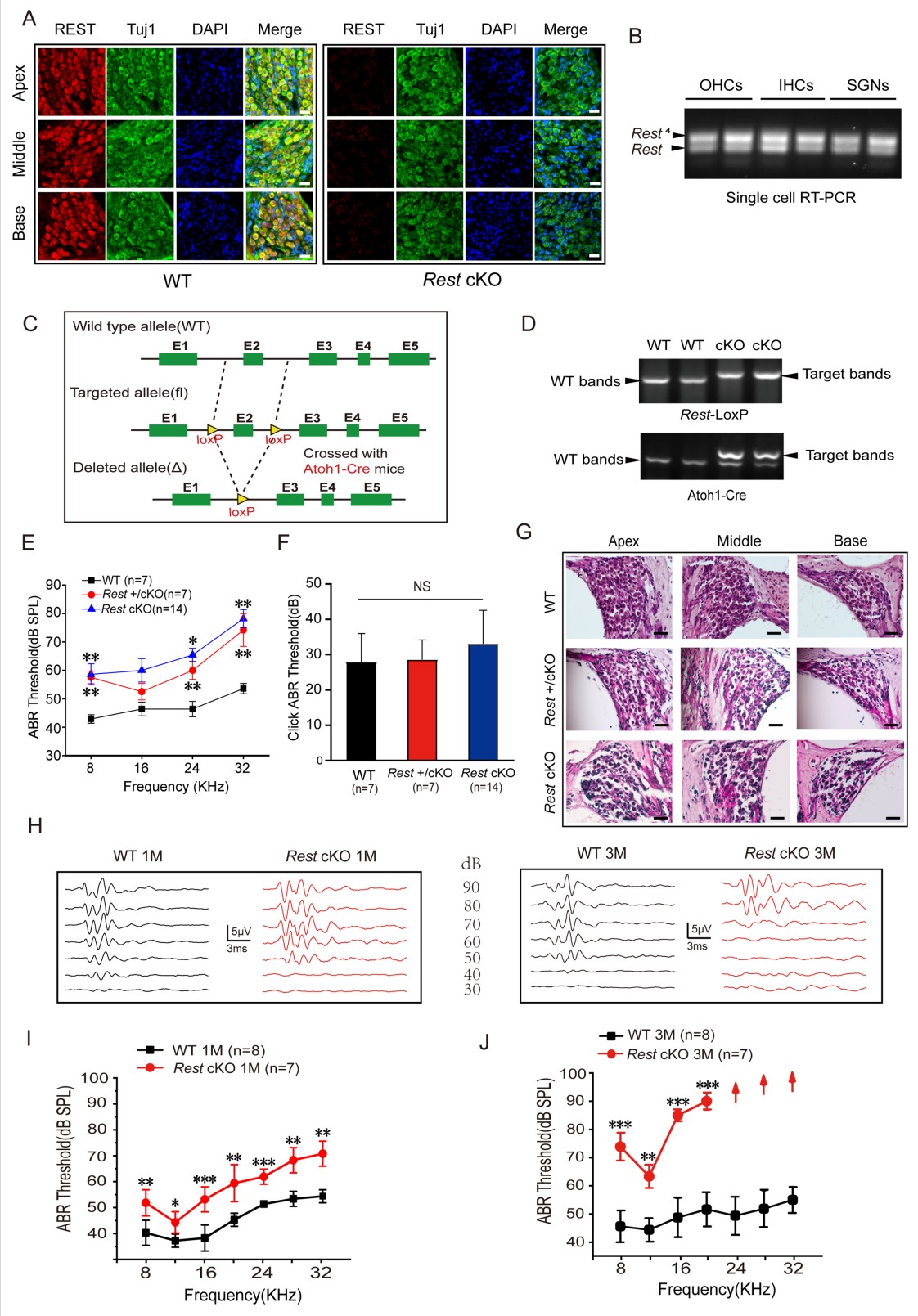

**Figure 1.** Repressor element 1-silencing transcription factor (REST) expression in the inner ear is essential for hearing. (**A**) Expression of REST in spiral ganglion neurons (SGNs) from the apical, middle, and basal cochlea of 1-month-old wild-type (WT) and *Rest* conditional knockout (cKO) mice. SGNs were stained using anti-REST (red) and anti-Tuj1, a neuron marker (green). The nuclei were stained with DAPI (blue). Scale bar: 20 μm. (**B**) Single-cell RT-PCR analysis of *Rest* expression in SGNs of WT mice at P40, outer hair cells (OHCs) at P13, and inner hair cells (IHCs) at P20. (**C**) Schematic diagram

*Figure 1 continued on next page*

*Figure 1 continued*

of *Rest* cKO generation process. Two LoxP sites were inserted into both alleles of REST, flanking the coding sequence of exon 2. Cre recombinase expression is activated by the specific *Atoh1* promoter in the cochlea and the exon 2 region between the homodromous LoxP sites, resulting in the loss of REST function cochlea. (**D**) PCR genotyping of WT and *Rest* cKO mice using genomic DNA prepared from tail biopsies. *Rest* cKO mice were identified by two-step PCR, first screening for the *Rest*-loxP target band and then further identifying for mice containing the *Atoh1-Cre* target band. (**E, F**) Auditory brainstem response (ABR) thresholds and click values were measured in WT, *Rest* +/cKO mice, and *Rest* cKO mice at 1–2 months old. (**G**) Morphometry changes in SGNs were observed in WT, *Rest* +/cKO, and *Rest* cKO mice. Scale bar: 50 μm. (**H**) Representative ABR waveforms in response to clicking (90–30 dB) sound pressure levels in 1- and 3-month-old WT and *Rest* cKO mice. (**I, J**) ABR threshold statistics of WT and *Rest* cKO mice at 1 month (**I**) and 3 months (**J**) of age in response to pure tone stimuli (8–32 kHz). Data are means ± SEM, **p<0.01, ***p<0.001.

The online version of this article includes the following source data and figure supplement(s) for figure 1:

**Source data 1.** Expression of REST in the inner ear and the hearing function in WT and *Rest* cKO mice.

**Figure supplement 1.** Identification of REST in HCs of WT and *Rest* cKO mice and hearing changes with age in *Rest* +/cKO mice.

**Figure supplement 1—source data 1.** Raw data of Single-cell RT-PCR and ABR from *Rest* +/cKO mice.

**Figure supplement 2.** Amplitudes and latencies of auditory brainstem response (ABR) waves Ⅰ and Ⅱ in 1- and 3-month-old wild-type (WT) and *Rest* conditional knockout (cKO) mice.

**Figure supplement 2—source data 1.** Data of ABR waveform analysis in 1- and 3-month-old WT and *Rest* cKO mice.

(P14), 63.2% (1-month-old), and 57.6% (3-month-old) of SGNs from *Rest* cKO mice failed to produce APs, indicating reduced excitability of apical SGNs in *Rest* cKO mice (*Figure 3B*).

SGNs at the base of the cochlea in P14, 1-, and 3-month-old WT mice could generate multiple AP patterns, including single AP (1 spike), phasic (2–6 spikes), and tonic (>6 spikes) (*Figure 3A*). In contrast, the proportions of SGNs generating multiple spikes decreased in *Rest* cKO mice, and many basal SGNs did not generate even a single AP. The ratios of these 'silent' neurons were 11.7, 35.3, and 32.8% in P14, 1-, and 3-month-old *Rest* cKO mice, respectively. These recordings indicate that the excitability of SGNs at the base of the cochlea in *Rest* cKO mice was reduced (*Figure 3B*). Additionally, the *Rest* cKO mice displayed hyperpolarized resting membrane potentials (RMP) in apical SGNs in 1- and 3-month-old mice and basal SGNs at P14 (*Figure 3C*). The AP thresholds in apical and basal SGNs were significantly increased in 3-month-old *Rest* cKO mice and apical SGNs at P14 compared to WT mice (*Figure 3D*). AP latencies were prolonged in SGNs of 1-month-old *Rest* cKO mice, and rheobase currents were increased in P14, 1-, and 3-month-old *Rest* cKO mice compared to WT mice (*Figure 3E and F*). Collectively, these data demonstrate that REST deficiency alters the electrophysiological properties of SGNs in mice, rendering the neurons less excitable.

## $K_v7.4$ is upregulated in the inner ear of *Rest* cKO mice

Ion channels are the key membrane proteins controlling neuronal excitability. They are responsible for setting the RMP and maintaining aspects such as the APs' duration, threshold, and firing frequency. Therefore, we explored which ion channels are involved in the reduced excitability of SGNs induced by REST deficiency. We focused on ion channels that are known to regulate the excitability of mouse SGNs (*Reijntjes and Pyott, 2016*) and contain RE-1 sites: key $Na_v$ and $K_v1$ subunits, $K_v4.3$, HCN1-2, and $K_v7$ (*Mucha et al., 2010*; *Reijntjes and Pyott, 2016*; *Uchida et al., 2010b*; *Figure 4A*). We compared the changes in the expression of these channels in the cochlea of WT and *Rest* cKO mice. As shown in *Figure 4A and B*, $K_v7.4$ channel mRNA and protein levels were significantly elevated in the cochlea of *Rest* cKO mice, while expression of other tested channels was not significantly affected. Further immunofluorescence results showed increased expression of $K_v7.4$ channel proteins in both HCs and SGNs (*Figure 4C–H*).

## Upregulation of $K_v7.4$ caused a decrease in SGNs' excitability in *Rest* cKO mice

$K_v7.4$ channels belong to the $K_v7$ channel family and are important components of $K_v7$ or 'M-type' neuronal $K^+$ currents. To determine whether increased $K_v7.4$ channel expression causes upregulation of channel function, we recorded $K_v7$ channel currents in the SGNs of WT and *Rest* cKO mice. SGNs were held at –20 mV, then hyperpolarized to –60 mV for 500 ms with a square voltage pulse before returning to holding potential. The deactivating $K^+$ current amplitude was calculated by subtracting the current amplitude measured at 10 ms before the end of the voltage step to –60 mV from the

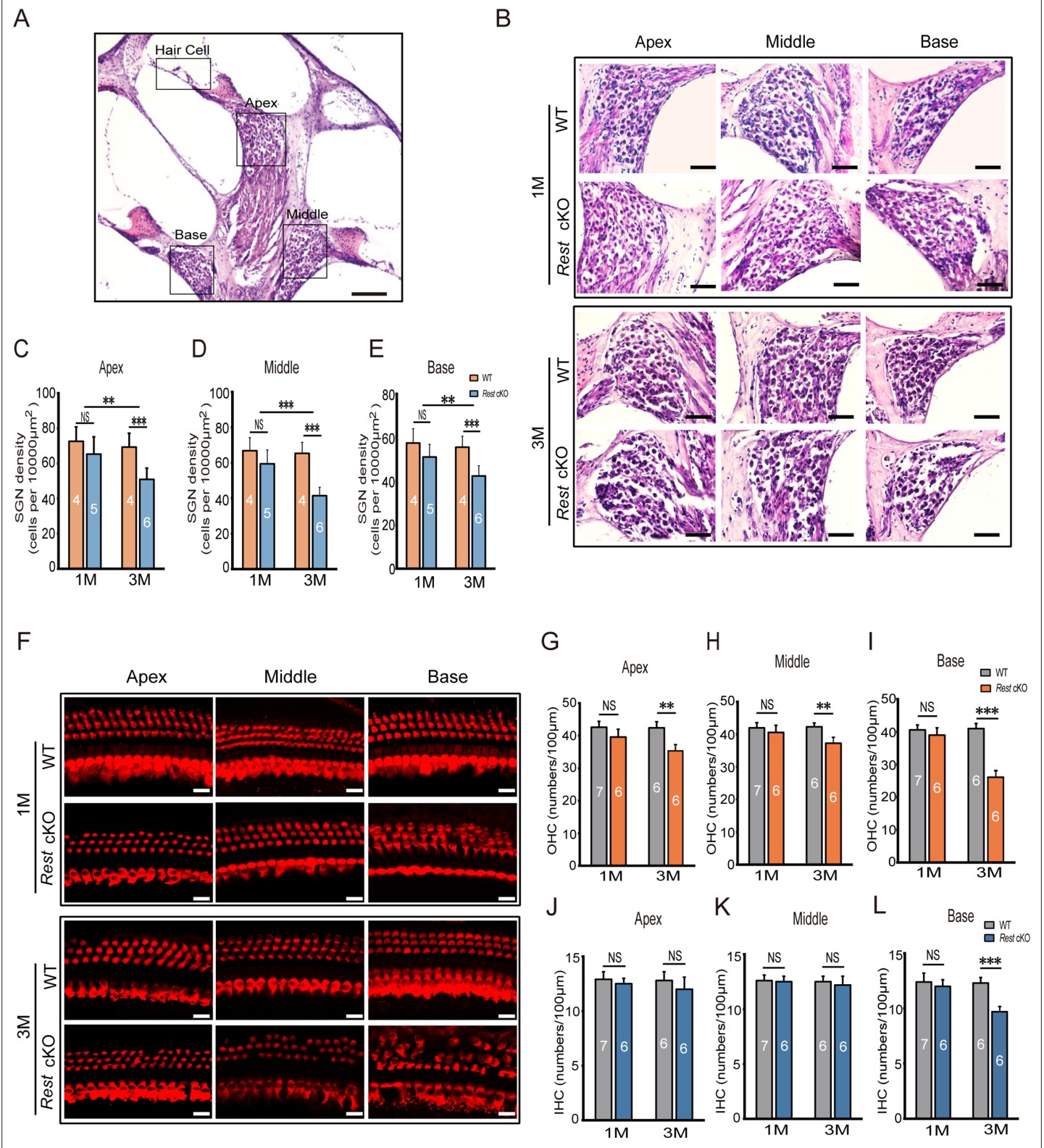

**Figure 2.** Degeneration of spiral ganglion neurons (SGNs) and hair cells (HCs) is observed in 3-month-old *Rest* conditional knockout (cKO) mice but not in 1-month-old mice. (**A, B**) Morphometry changes in SGNs were observed in the apical, middle, and basal cochlea of 1- and 3-month-old wild-type (WT) and *Rest* cKO mice. Scale bar: 200 µm in (**A**) and 50 µm in (**B**). (**C–E**) SGNs were quantified in all three regions of the cochlea. (**F**) Myo7a-stained HCs from 1- and 3-month-old WT and *Rest* cKO mice. Scale bar: 20 µm. (**G–I**) Quantification of outer hair cells (OHCs) at the cochlea's apical middle

*Figure 2 continued on next page*

*Figure 2 continued*

and basal regions in WT and *Rest* cKO mice (1M and 3M). (**J–L**) Quantification data of inner hair cells (IHCs) in WT and *Rest* cKO mice. Data are means ± SEM, \*\*p<0.01, \*\*\*p<0.001.

The online version of this article includes the following source data and figure supplement(s) for figure 2:

**Source data 1.** Original images and quantitative data of SGNs and HCs in WT and *Rest* cKO mice.

**Figure supplement 1.** There are no detectable alterations in hair cell (HC) and spiral ganglion neuron (SGN) morphology in *Rest* conditional knockout (cKO) mice at P1, P7, and P14.

**Figure supplement 1—source data 1.** Original images of SGNs and HCs in WT and *Rest* cKO mice at P1,P7 and P14.

current amplitude measured at 10 ms after the onset of this voltage step, as 'a-b' or 'c-d' shown in *Figure 5A1*. The K$_v$7 currents were identified as the deactivation current component, blocked by the K$_v$7 channel blocker XE991 (3 µM), which was calculated as (a-b) – (c-d) (*Figure 5A1*). The K$^+$ currents recorded from SGN often displayed a degree of rundown. Thus, the K$_v$7 current component in these recordings may have been underestimated.

Representative K$_v$7 currents and time course of K$^+$ currents inhibited by XE991 recorded in SGNs of P14, 1-, and 3-month-old WT mice are shown in *Figure 5A and B*, respectively. The presence of XE991-sensitive currents (K$_v$7 currents) in SGNs was demonstrated by measuring the K$^+$ current density of individual SGNs at –20 mV before (basal) and after the application of XE991 (3 µM) (*Figure 5C*). We found no significant upregulation of K$_v$7 current density in the SGNs of P14 mice compared to that in WT mice, while K$_v$7 current density was dramatically increased in the SGNs at the apex and base of 1- and 3-month-old *Rest* cKO mice (*Figure 5D and E*). The capacitances of SGNs from P14, 1-, and 3-month-old WT mice were 14.47 ± 4.33 pF, 15.09 ± 5.31 pF, and 15.19 ± 5.32 pF, and capacitances of SGNs from P14, 1-, and 3-month-old *Rest* cKO mice were 14.37 ± 4.62 pF, 14.94 ± 4.74 pF, and 15.24 ± 3.33 pF.

K$_v$7.2 and K$_v$7.3 also contribute to K$_v$7 channel currents in many neurons, which are also expressed in SGNs (*Jin et al., 2009*). To determine whether K$_v$7.2 and K$_v$7.3 are involved in the upregulation of K$_v$7 channel currents, we examined the mRNA levels of K$_v$7.2 and K$_v$7.3 in the cochlea of WT and *Rest* cKO mice using real-time PCR. As shown in *Figure 4A*, the expression levels of K$_v$7.2 and K$_v$7.3 were not significantly different between WT and *Rest* cKO mice. Further immunofluorescence staining confirmed that the expression of K$_v$7.2 and K$_v$7.3 channels remained unchanged in the SGNs of *Rest* cKO mice (*Figure 5—figure supplement 1*). These data indicate that deletion of cochlear REST upregulates K$_v$7.4 channels rather than K$_v$7.2 and K$_v$7.3 in SGNs. The results were unexpected as deletion of REST was shown to upregulate K$_v$7.2 in mouse sensory neurons (*Zhang et al., 2019*), possibly suggesting that additional factors contribute to cell-type-specific regulation of K$_v$7 channels by REST.

To further confirm the inhibitory effect of REST on K$_v$7.4, we overexpressed REST in CHO cells stably expressing K$_v$7.4 channels and observed a significant decrease in K$_v$7.4 current density (*Figure 5—figure supplement 2*). The Western blot results confirmed that overexpression of REST significantly reduced the protein expression level of K$_v$7.4 channels in CHO cells (*Figure 5—figure supplement 2*), indicating that REST is responsible for the reduced K$_v$7.4 currents by inhibiting K$_v$7.4 protein expression.

## Upregulation of K$_v$7.4 results in dysfunction of OHCs in *Rest* cKO mice

K$_v$7.4 is the only K$_v$7 family member reported to be expressed in OHCs so far and is necessary to maintain the electrophysiological properties of OHCs and their functions (*Johnson et al., 2011*; *Marcotti and Kros, 1999*; *Perez-Flores et al., 2020*). We first recorded K$_v$7 currents in the OHCs of P10–14 WT and *Rest* cKO mice. Whole-cell K$^+$ currents were recorded from OHCs, and the XE991-sensitive currents denoted hereafter as K$_v$7-mediated currents are shown in *Figure 6A–E*. As with SGNs, K$_v$7 channel currents were significantly increased in the OHCs of *Rest* cKO mice (*Figure 6F*).

Subsequently, we measured the distortion product otoacoustic emissions (DPOAEs). DPOAEs depend on active processes in the cochlea, which are linked to the normal function of OHCs (*Dallos, 1992*). DPOAEs detect low-level sounds produced by functional OHCs and emitted back to the ear canal. We demonstrated that DPOAE thresholds were significantly increased in *Rest* cKO mice (*Figure 6G and H*), indicating a decline in the function of OHCs; such an effect is consistent with the upregulation of K$_v$7.4 in OHCs.

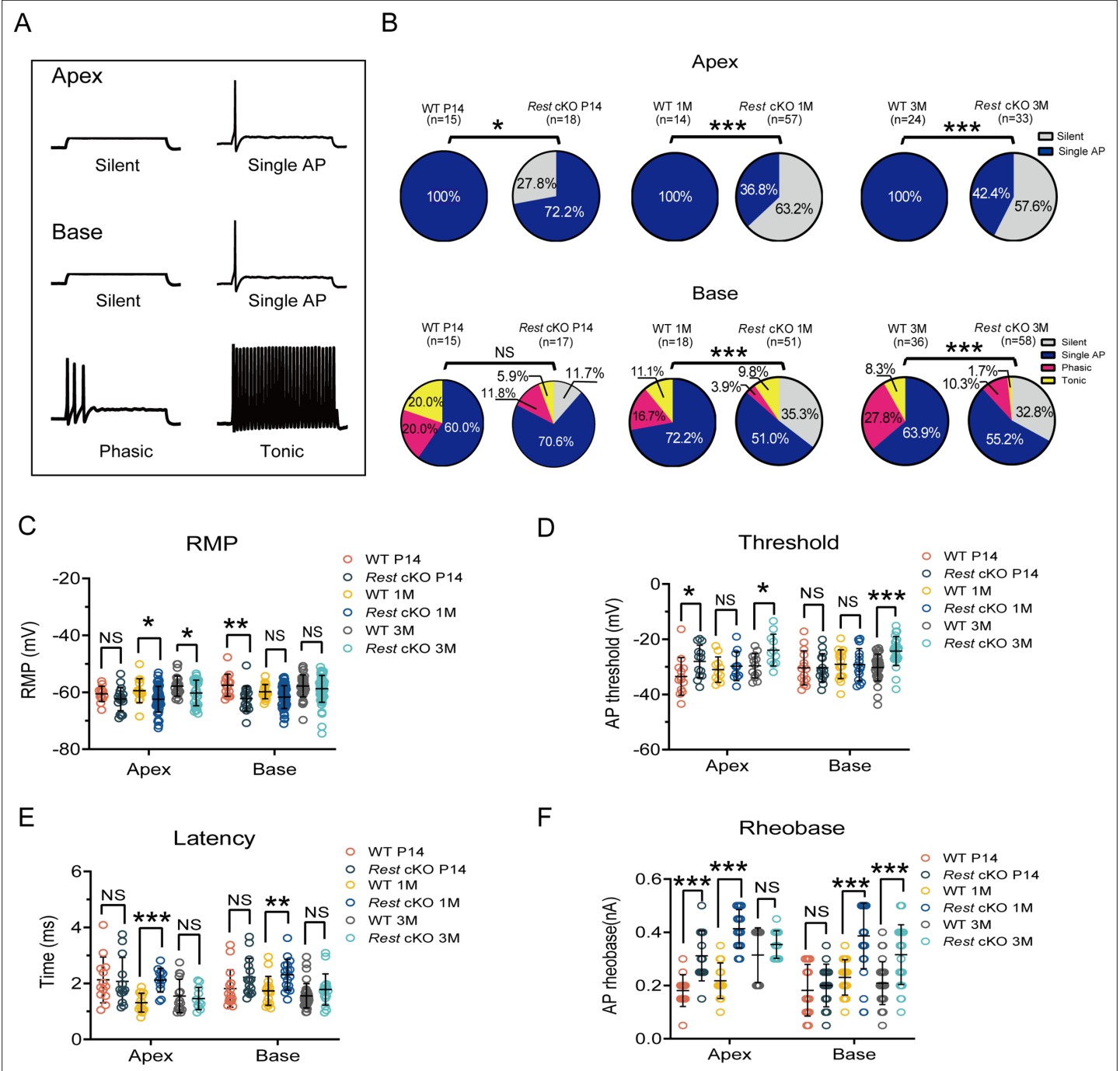

**Figure 3.** Reduced excitability of spiral ganglion neurons (SGNs) in *Rest* conditional knockout (cKO) mice. (**A**) Representative traces show spike patterns of SGNs in the apical and basal cochlea, recorded with whole-cell patch clamps. Spikes were generated by 0.4 nA current injection. (**B**) Pie charts illustrating the percentage abundance of the different spike patterns of SGNs in wild-type (WT) and *Rest* cKO mice. (**C–F**) The resting membrane potentials (RMPs) (**C**), action potential thresholds (**D**), latencies (**E**), and rheobases (**F**) were recorded from apical and basal SGNs in WT and *Rest* cKO mice at P14, 1, and 3 months. Data are means ± SEM, *p<0.05, **p<0.01,***p<0.001.

The online version of this article includes the following source data for figure 3:

**Source data 1.** Data of excitability of SGNs in WT and *Rest* cKO mice.

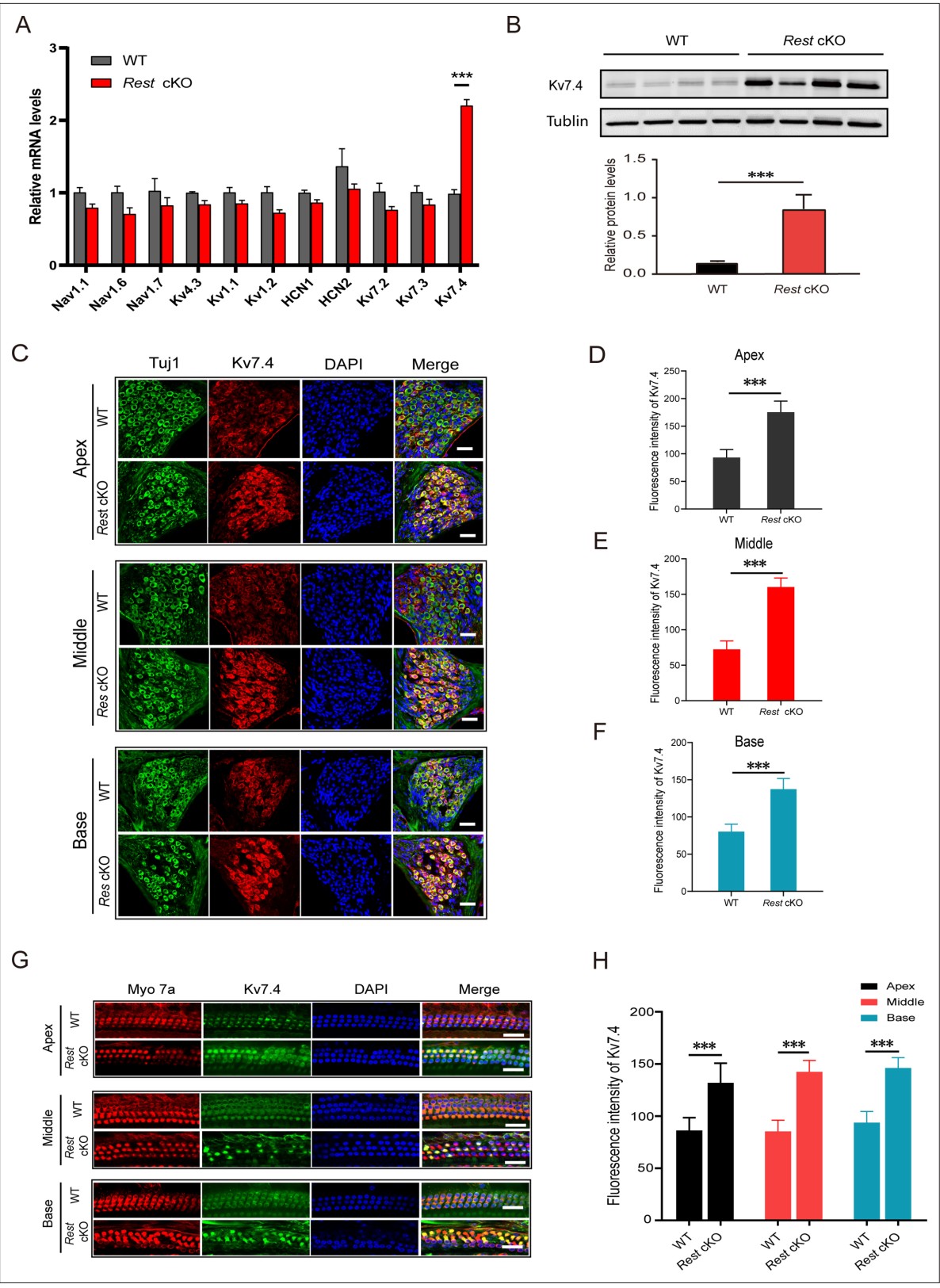

**Figure 4.** K$_v$7.4 expression is increased in spiral ganglion neurons (SGNs) and hair cells (HCs) of *Rest* conditional knockout (cKO) mice. (**A, B**) *Kcnq4* expression is significantly increased in the cochlea of *Rest* cKO mice at the mRNA and protein (K$_v$7.4) levels as determined by real-time PCR (**A**) and Western blotting (**B**). (**C–F**) Immunofluorescence shows increased K$_v$7.4 (red) expression in the SGNs from apical, middle, and basal cochlea in *Rest* cKO mice. Scale bar: 20 μm in (**C**, **G**). (**G, H**) K$_v$7.4 expression increases in HCs of *Rest* cKO mice. Data are means ± SEM, ***p<0.001.

*Figure 4 continued on next page*

*Figure 4 continued*

The online version of this article includes the following source data for figure 4:

**Source data 1.** Data for Kv7.4 expression in SGNs and OHCs in WT and *Rest* cKO mice.

## Pharmacological activation of K$_v$7 channels impairs hearing in mice

To verify that upregulation of K$_v$7.4 channel functions results in hearing loss in vivo, we administered fasudil (i.p., 10 mg/kg and 20 mg/kg, every 2 days) to WT mice and evaluated their hearing function. Fasudil is a Rho-related protein kinase (ROCK) inhibitor (*Shi and Wei, 2013*), which was recently shown to directly and selectively (amongst other K$_v$7 subunits) activate K$_v$7.4-containing channels (*Zhang et al., 2016*). ABR was measured before and 14 and 28 days after fasudil administration (*Figure 7A*). The hearing thresholds of mice in the 10 mg/kg and 20 mg/kg fasudil-treated groups were significantly higher on day 28, indicating hearing impairment (*Figure 7B*). Analysis of the amplitudes and latencies of ABR waves I and II revealed shorter amplitudes with click and pure-tone stimulation and longer latencies with pure-tone stimulation after fasudil treatment, suggesting that fasudil may trigger auditory nerve processing dysfunction at the level of HCs and SGNs (*Figure 7—figure supplement 1*).

We subsequently determined whether the loss of SGNs or HCs induced hearing impairment. We demonstrated that all groups of mice displayed normal cochlear morphology with no loss of SGNs and HCs (*Figure 7—figure supplement 2*), suggesting that fasudil treatment did not alter the morphology or number of HCs and SGNs.

Next, we assessed the effect of fasudil on cochlear cell function. We examined the effects of fasudil on K$_v$7 currents in SGNs and their excitability. K$_v$7 current density in SGNs in both the apex and base was significantly increased in the fasudil-treated groups (*Figure 7C and D*). Additionally, SGN excitability was dramatically reduced (*Figure 7E*), manifesting as hyperpolarized RMPs, elevated AP thresholds, prolonged AP latencies, and elevated AP rheobases (*Figure 7F–I*).

## K$_v$7 channel blocker XE991 rescues hearing loss in *Rest* cKO mice

To demonstrate that the upregulation of K$_v$7.4 channels caused by REST deficiency leads to deafness in mice, we investigated whether blocking K$_v$7.4 channels could rescue hearing loss in *Rest* cKO mice. We tested the hearing function of *Rest* cKO mice before and after administering XE991, a K$_v$7 channel inhibitor (*Figure 8A*) and found that the ABR thresholds of *Rest* cKO mice were significantly reduced following treatment with 0.25 mg/kg XE991. We further observed an increase in amplitude and a decrease in latencies of ABR waves I and II, indicating an improvement in hearing (*Figure 8B and C*, *Figure 8—figure supplement 1A–D*). Moreover, the ABR thresholds of mice in the XE991 0.5 mg/kg treatment group were also significantly lower at low frequencies (4–8 kHz), with no differences in the amplitude and latency of waves I and II, indicating a partial improvement in hearing (*Figure 8B and C*, *Figure 8—figure supplement 1E–H*). Reduction of XE991 efficacy at a higher dose may mean additional on- or off-target side effects of the drug at a higher dosage.

## Discussion

REST, a transcriptional repressor, plays a dominant role in neural development and differentiation. Emerging evidence indicates that REST is an important player in the developed nervous system, including recognized roles in chronic pain development (*Rose et al., 2011*; *Uchida et al., 2010a*, *Zhang et al., 2018*; *Zhang et al., 2019*) and aging (*Cheng et al., 2022*; *Lu et al., 2014*; *Zullo et al., 2019*). However, the role of REST in hearing is still unknown. Here, we generated conditional inactivation of *Rest* in the cochlea. We found that specific deletion of *Rest* resulted in enhanced K$_v$7.4 channel expression in HCs and SGNs and increased K$_v$7 currents in both cell types, leading to reduced SGN excitability and compromised HC function, with both effects resulting in progressive hearing loss in mice. Consistently, administration of the compound with K$_v$7.4 channel activator activity, fasudil, caused hearing loss in WT mice, while the K$_v$7 channel inhibitor XE991 modestly rescued the hearing deficit of *Rest* cKO mice.

Several loss-of-function mutations of K$_v$7.4 in humans and animal models were reported to cause gradual degeneration of HCs and SGNs, resulting in an autosomal-dominant form of non-syndromic progressive hearing loss (DFNA2) (*Coucke et al., 1999*; *Jentsch et al., 1990*; *Kharkovets et al.,*

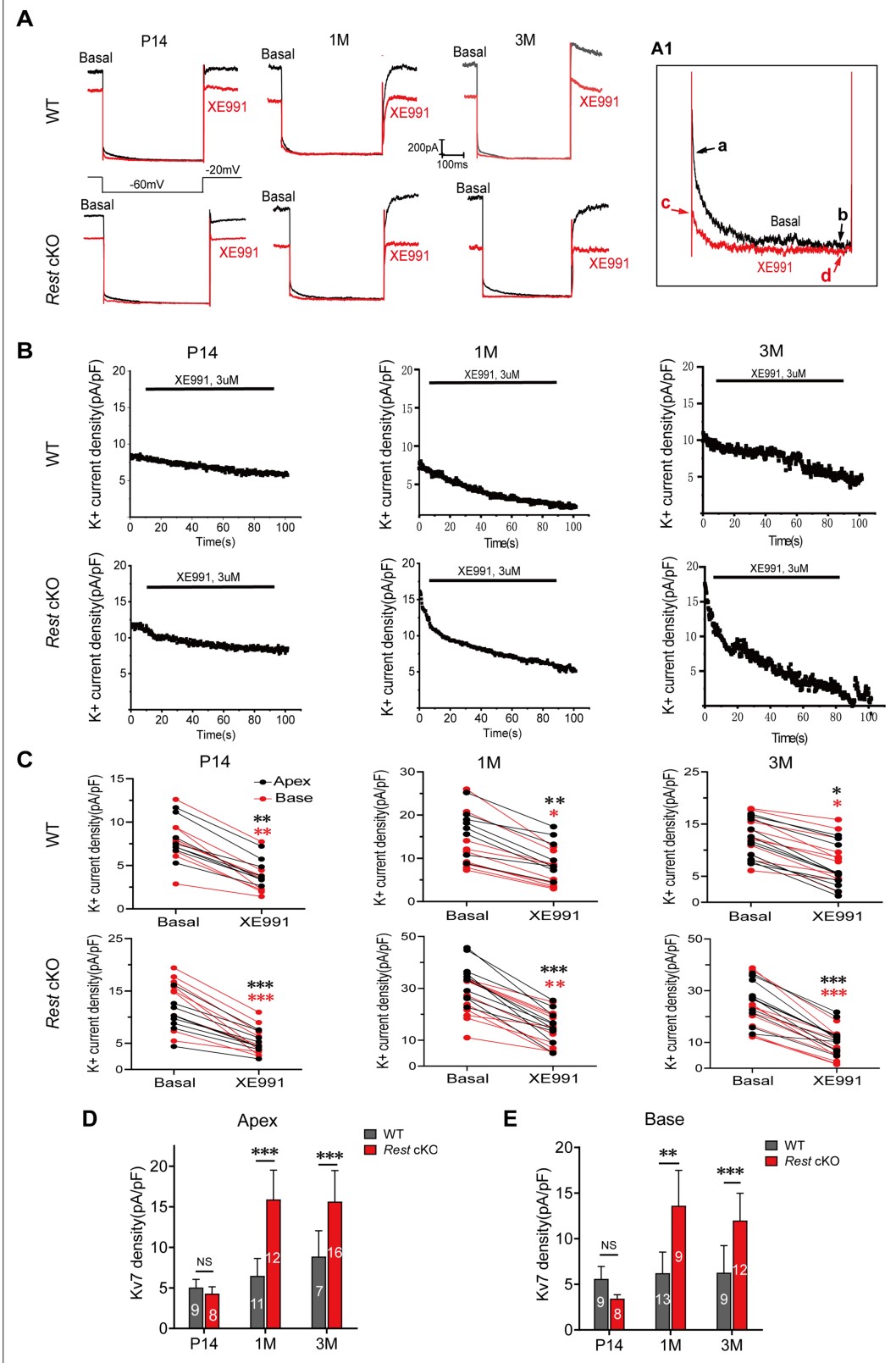

**Figure 5.** Increased K$_v$7 currents in the spiral ganglion neurons (SGNs) of *Rest* conditional knockout (cKO) mice. (**A**) Whole-cell K$_v$7 currents were recorded in wild-type (WT) and *Rest* cKO SGNs. Representative K$_v$7 currents were calculated as XE991-sensitive currents upon voltage step from –20 mV to –60 mV. Arrowheads indicate the points at which XE99-sensitive currents were measured (**A1**). (**B**) Time course of outward K$^+$ current inhibition by XE991

*Figure 5 continued on next page*

*Figure 5 continued*

(3 µM). (**C**) Comparison of outward K$^+$ current at –20 mV in individual SGNs from the apex (black) and base (red) of the cochlea in the absence (basal) and presence of XE991 (3 µM). (**D, E**) Summary data show that XE991-sensitive currents (K$_v$7 current) density is significantly increased in the apical and basal SGNs of 1- and 3-month-old *Rest* cKO mice compared to WT mice. Data are means ± SEM, *p<0.05, **p<0.01, ***p<0.001.

The online version of this article includes the following source data and figure supplement(s) for figure 5:

**Source data 1.** Data of Kv7.4 channel currents recorded in SGNs of WT and *Rest* cKO mice.

**Figure supplement 1.** Expression of K$_v$7.2 and K$_v$7.3 in the spiral ganglion neurons (SGNs) of wild-type (WT) and *Rest* conditional knockout (cKO) mice.

**Figure supplement 1—source data 1.** Images and quantitative data of Kv7.2 and Kv7.3 expression in SGNs of WT and *Rest* cKO mice.

**Figure supplement 2.** Repressor element 1-silencing transcription factor (REST) inhibited K$_v$7.4 channels in K$_v$7.4-transfected Chinese hamster ovary (CHO) cells.

**Figure supplement 2—source data 1.** Voltage clamp and Western Blotting data of Kv7.4 inhibited by REST in CHO cells.

---

*2006*; *Kim et al., 2011*; *Kubisch et al., 1999*). Our results suggest that 1-month-old *Rest* cKO mice showed no degeneration of SGNs and HCs, which indicates that enhanced K$_v$7.4 expression/current can also lead to hearing loss. Our findings reinforce a hypothesis whereby a specific range of basal K$_v$7.4 functional expression is necessary for good hearing. A reduction or increase outside of this safety range causes cochlear dysfunction. It is currently unclear whether the upregulation of K$_v$7.4 and degeneration of SGNs and HCs, observed in 3-month-old *Rest* cKO mice, are related phenomena; this will require further investigation. Importantly, we identified the transcriptional suppressor REST as an essential regulator of hearing in mice.

The mammalian *Rest* gene is 20–30 kb long and consists of five exons and four introns. *Rest* pre-mRNA can generate multiple splice variants with alternative splicing. *Rest*[4], a result of the alternative splicing of exon 4, was reported to reduce the repressive activity of REST, which is present in the HCs of P1–P7 mice (*Nakano et al., 2018*). Heterozygous deletion of exon 4 resulted in the gain of function of REST, leading to HC degeneration and deafness, which is associated with DFNA27 hearing loss (*Nakano et al., 2018*), suggesting that *Rest*[4] is necessary for HC development and hearing.

Here, we generated cochlea-specific *Rest* knockout mice using *Atoh1-Cre* mice crossed with *Rest* exon 2-flox mice to determine whether REST deficiency affects hearing. Because exon 2 is an exon essential for a functional REST protein, its knockout results in the loss of function of the full-length and all alternatively spliced REST variants. Hence, our results demonstrate that a total loss of functional REST in the cochlea causes hearing loss, suggesting that a full-length REST is vital for HC and SGN functions. REST was first identified in the chick cochlea by RT-PCR in 2002 (*Roberson et al., 2002*). The REST splice variant was recently demonstrated in immature HCs in mice (*Nakano et al., 2018*). Our study found that REST is expressed in adult HCs and immature and mature SGNs.

K$_v$7.4 channel is a voltage-dependent K$^+$ channel encoded by the *KCNQ4* gene. It belongs to the K$_v$7 K$^+$ channel family containing five K$_v$ channels, K$_v$7.1–K$_v$7.5 (*Jentsch, 2000*; *Jones et al., 2021*). The K$_v$7 channels are widely distributed in the neurons, heart, vascular smooth muscle cells, and cochlea (*Brueggemann et al., 2014*; *Delmas and Brown, 2005*; *Jentsch, 2000*; *Kharkovets et al., 2000*; *Neyroud et al., 1997*). In the cochlea, K$_v$7.4 is mainly expressed in the OHCs, IHCs, and SGNs (*Beisel et al., 2005*; *Dierich et al., 2020*; *Kharkovets et al., 2000*). When expressed in heterologous expression systems, K$_v$7.4 channels conduct slowly activating and deactivating currents that do not inactivate (i.e., as shown in *Figure 5—figure supplement 2*). We noted that the XE991-sensitive currents recorded from OHCs had faster kinetics than those expected from heterologously expressed K$_v$7.4 homo-tetramers (*Figure 6E*). Hence, we suggest that OHCs may contain endogenous factors or modulators that alter the K$_v$7.4 channel kinetics. On the other hand, currents recorded from SGNs had an 'M-like' slow-kinetics component, especially in neurons from *Rest* cKO mice (*Figure 5*). Loss-of-function mutations in *KCNQ4* cause DFNA2 sensorineural deafness (*Coucke et al., 1999*; *Kharkovets et al., 2006*). Our results demonstrate that upregulation of K$_v$7.4 channels also leads to hearing loss in mice, presumably by silencing SGNs and inducing dysfunction in HCs.

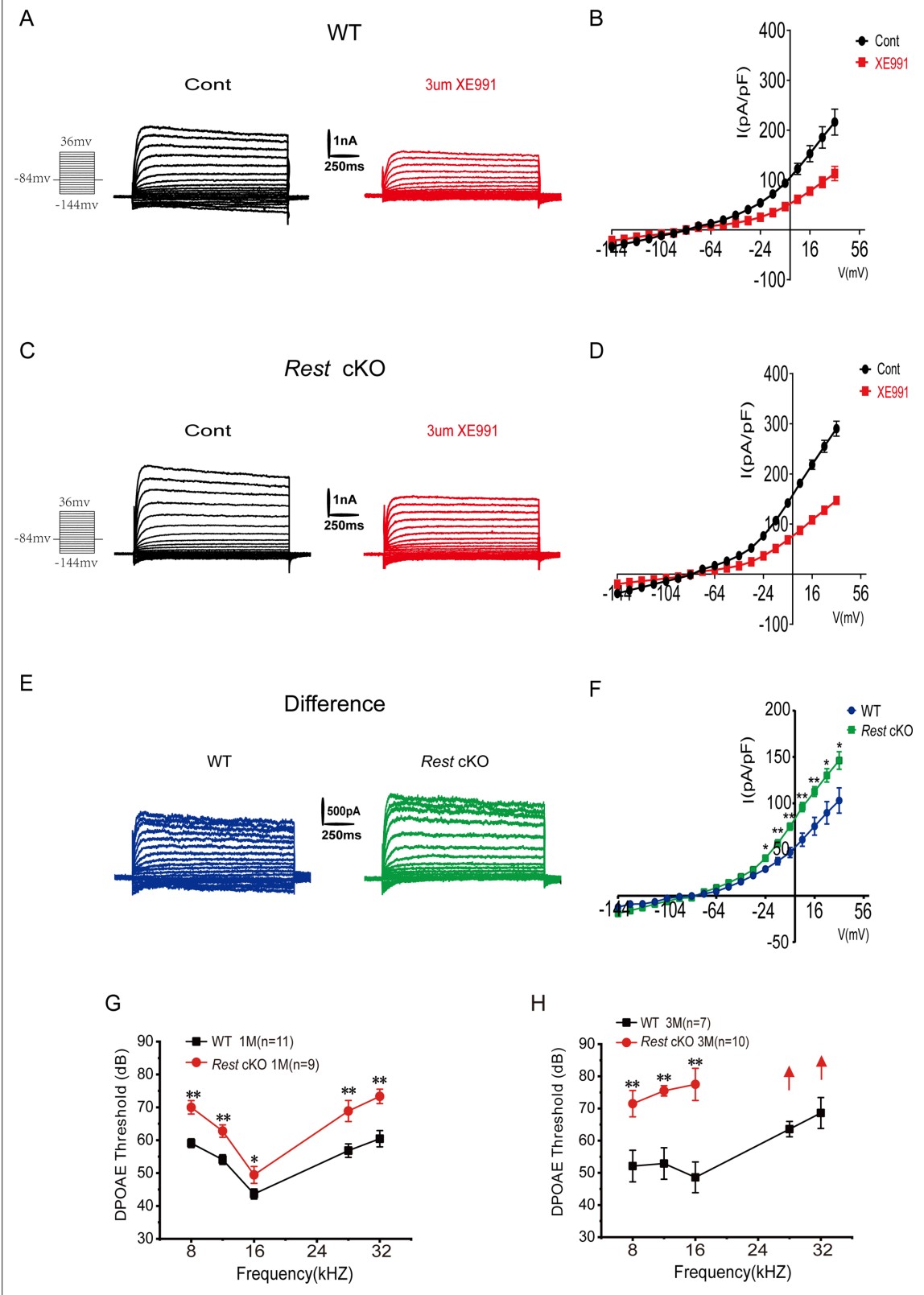

**Figure 6.** Increased K$_v$7 currents in the outer hair cells (OHCs) of *Rest* conditional knockout (cKO) mice. (**A**) Representative current traces recorded from the OHCs of P13 wild-type (WT) mice and the resulting traces after application of XE991. K$^+$ currents were obtained from a holding potential of –84 mV and stepped up from –144 to +34 mV, with 10 mV increments (left panel). (**B**) Current density–voltage curves are shown before and after XE991 (3 μM) application in WT mice. (**C**) Similar data was obtained from the OHCs of *Rest* cKO mice in the control condition, and after XE991 (3 μM), current traces

*Figure 6 continued on next page*

*Figure 6 continued*

were applied. (**D**) Current density–voltage curves are shown before and after the application of XE991 in *Rest* cKO mice. (**E**) The difference current traces, which represent the XE991-sensitive currents. (**F**) Difference current density–voltage curve obtained from OHCs in WT and *Rest* cKO mice. (**G, H**) Distortion product otoacoustic emission (DPOAE) threshold measurement of 1- and 3-month-old WT and *Rest* cKO mice. Data are means ± SEM, *p<0.05, **p<0.01.

The online version of this article includes the following source data for figure 6:

**Source data 1.** Data of Kv7.4 channel currents recorded in OHCs of WT and *Rest* cKO mice.

In this study, we did not directly measure the binding of REST to *Kcnq4,* but direct binding has been reported before (*Iannotti et al., 2013*). REST regulates the expression of a variety of ion channel genes, including genes coding for HCN channels (*Kuwahara, 2013*; *McClelland et al., 2011*), $Na_v1.2$ channels (*Armisén et al., 2002*; *Pozzi et al., 2013*), $K_v7.2/K_v7.3$ channels (*Mucha et al., 2010*; *Rose et al., 2011*), and T-type $Ca^{2+}$ channels (*van Loo et al., 2012*). Although some ion channel-related REST target genes were also expressed in the cochlea, our results showed that REST deletion only significantly upregulated $K_v7.4$ channels in SGNs and HCs (amongst genes tested here, see *Figure 4A*). REST deletion specifically upregulates *Kcnq4* but not other RE1-containing ion channel-related genes in the cochlea that remain to be investigated. However, it is worth noting that different RE1 sites bind REST with different affinities (*Ooi and Wood, 2007*); hence, at least some potentially REST-targeted genes may not be tonically suppressed (and, thus, not affected by REST deletion). In addition, REST could suppress additional regulators of other genes. Therefore, the outcome of REST deletion may not necessarily affect the expression of all RE1-containing genes in the same way. Moreover, the genes involved may be cell- and tissue-specific.

To verify that upregulation of $K_v7.4$ channels can cause hearing loss, we treated mice with the compound with $K_v7.4$ channel activator activity, fasudil. Although fasudil was initially described as a ROCK inhibitor (*Shi and Wei, 2013*), recently, *Li et al., 2017* found that fasudil selectively increased $K_v7.4$ channel currents in dopaminergic (DA) neurons in the ventral tegmental area (VTA). Moreover, Zhang et al. revealed that fasudil selectively activated $K_v7.4$ channels in vascular smooth muscles (*Zhang et al., 2016*). It was further demonstrated that fasudil has a selective direct opening effect on $K_v7.4$ channels without significantly affecting other members of the $K_v7$ channel family (*Zhang et al., 2016*). We observed increased $K_v7$ channel currents in SGNs in mice 28 days after recurrent fasudil administration, accompanied by deafness.

It should be noted that $K_v7.4$ channels are also expressed in neurons of the dorsal nucleus of the cochlea and the hypothalamus nucleus in the central auditory pathway (*Kharkovets et al., 2000*). Thus, fasudil could influence hearing by affecting $K_v7.4$ channels in these neurons. The amplitude and latency of ABR waves, which represent the process of neural transmission in the auditory system, were measured to elucidate further the exact site of action of systemic fasudil on hearing. Fasudil administration alters waves I and II of the ABR characteristics, indicating the involvement of HCs and SGNs (*Figure 7—figure supplement 1*). The actions of fasudil are reversible upon acute application (*Zhang et al., 2016*). Under the current conditions where the drug was administered chronically, we conceive that long-term and irreversible effects may ensue.

XE991, a $K_v7$ channel blocker, was used to demonstrate that upregulation of $K_v7.4$ channels caused by REST defects is involved in deafness in mice. Our results indicated that at low concentrations (0.25 mg/kg), XE991 significantly rescued the hearing of *Rest* cKO mice, while at high concentrations (0.5 mg/kg), XE991 had a more modest effect on restoring the hearing of these mice. As a certain number of $K_v7.4$ channels is necessary to regulate cell excitability and thus maintain hearing in mice, their up- or downregulation beyond the basal range of expression would impact hearing. Therefore, we hypothesize that 0.25 mg/kg XE991 could appropriately inhibit the overexpressed $K_v7.4$ channels in *Rest* cKO mice, restoring them to a 'safe' range. In contrast, 0.5 mg/kg XE991 may overinhibit $K_v7.4$ channels beyond the scope in which $K_v7.4$ channels perform their normal function and ultimately fail to improve hearing in mice. Other on- or off-target side effects of higher doses of XE991 also cannot be excluded.

In conclusion, our findings revealed that a critical homeostatic level of functional expression of the $K_v7.4$ channel is required to maintain proper auditory functions. REST is necessary for hearing through regulating such functional expression of $K_v7.4$. The study contributes to an understanding

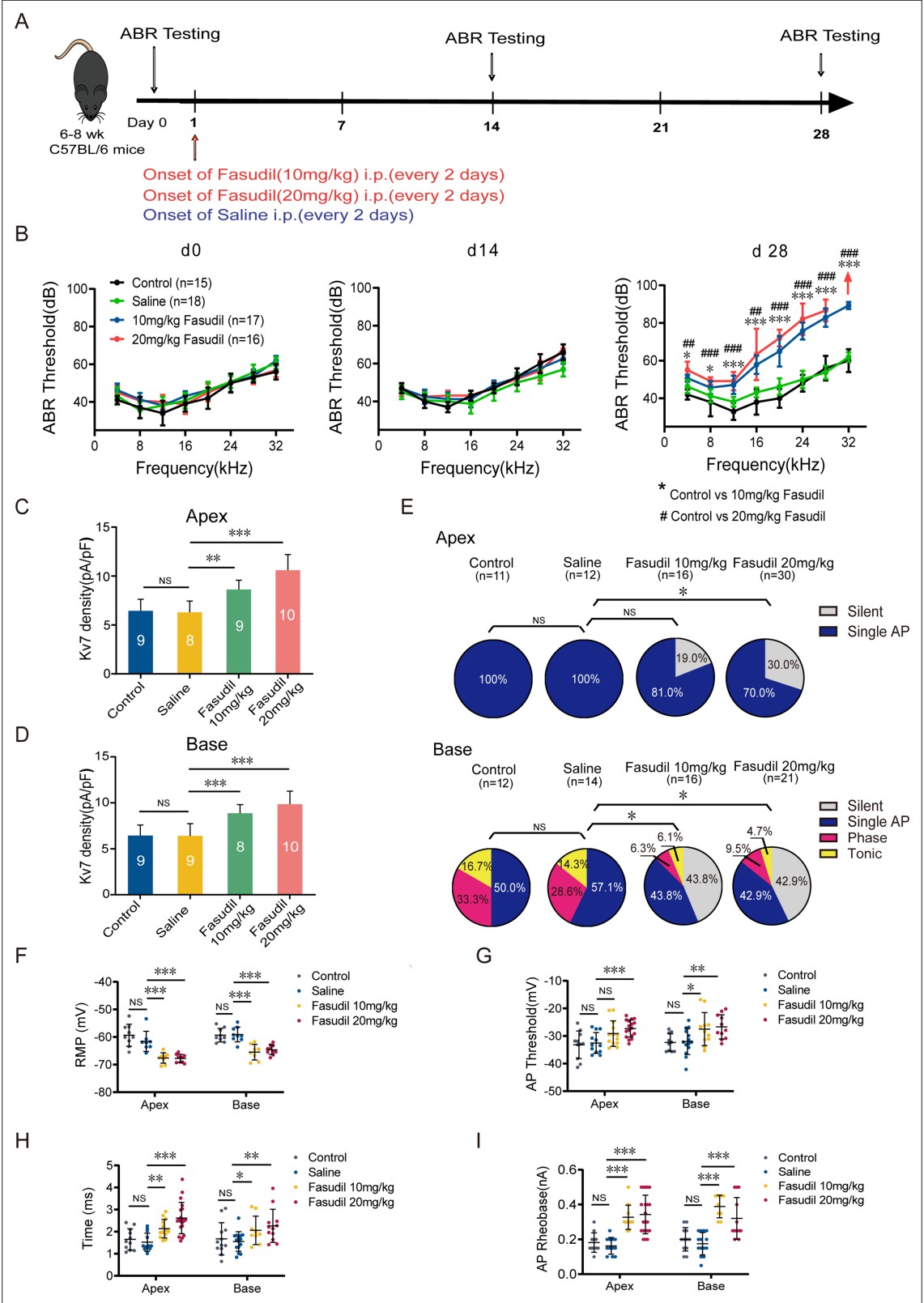

**Figure 7.** K$_v$7.4 channel activation induced hearing impairments. (**A**) Timeline for auditory brainstem response (ABR) testing and drug treatment. Fasudil was administered to 6–8-week-old wild-type (WT) mice intraperitoneally at 10 mg/kg or 20 mg/kg every other day for 28 days. (**B**) ABR thresholds in control, saline, and fasudil (10 mg/kg, 20 mg/kg) treatment groups on days 0, 14, and 28. (**C, D**) Summary data show increased K$_v$7 current density in the apical and basal spiral ganglion neurons (SGNs) of fasudil (10 mg/kg, 20 mg/kg)-treated mice. (**E**) Summary of spike patterns recorded in SGNs from

*Figure 7 continued on next page*

*Figure 7 continued*

control, saline, and fasudil (10 mg/kg, 20 mg/kg)-treated mice. (**F–I**) Resting membrane potentials (RMPs), action potential thresholds, latencies, and rheobases were recorded from mice's apical and basal SGNs. Data are means ± SEM, *p<0.05, **p<0.01, ***p<0.001.

The online version of this article includes the following source data and figure supplement(s) for figure 7:

**Source data 1.** Data of the effect of fasudil on hearing function and excitability of SGNs in WT mice.

**Figure supplement 1.** Amplitudes and latencies of auditory brainstem response (ABR) waves Ⅰ and Ⅱ in control, saline, and fasudil-treated mice.

**Figure supplement 1—source data 1.** Data of latencies and amplitudes of ABR waves in fasudil-treated mice.

**Figure supplement 2.** Fasudil did not alter the morphometry of spiral ganglion neurons (SGNs) and hair cells (HCs).

**Figure supplement 2—source data 1.** Morphological data of cochlear cells in fasudil-treated mice.

of the essential functions of REST in the auditory system and provides new insight into the potential future treatments of hearing disorders associated with the $K_v7.4$ channel.

## Methods

### Animals

All experimental animal protocols were performed following the Animal Care and Ethical Committee of Hebei Medical University (Shijiazhuang, China). The mice were bred and housed under a 12:12 light–dark cycle. To generate *Atoh1-Cre: Rest^flox/flox* conditional knockout mice (*Rest* cKO), *Rest^flox/flox* mice (*Soldati et al., 2012*) were crossed with *Atoh1-Cre* mice (Jackson Laboratories, Bar Harbor, ME, stock no. 011104). Genotyping of the mouse tails and cochlear tissues was performed using the primers listed in *Source data 1*. All experimental animal protocols were performed following the Animal Care and Ethical Committee of Hebei Medical University (Shijiazhuang, China; 01644).

### Auditory Brainstem Responses (ABRs) and Distortion Product Otoacoustic Emissions (DPOAE)

ABR measurements were performed as previously described (*Shen et al., 2018*). Briefly, mice were anesthetized with ketamine (100 mg/kg) and xylazine (10 mg/kg). Three electrodes were inserted subcutaneously at the vertex of the head (reference), ipsilateral mastoid (recording), and contralateral rear leg of the mice (ground). At 8, 12, 16, 20, 24, 28, and 32 kHz, click or tone stimuli were emitted from 20 dB with an attenuation interval of 5 dB. ABRs were measured using a System III workstation (Tucker Davis Technologies, Alachua, FL) in an IAC BioSigRP Sound booth (GM Instruments, Irvine, UK). The hearing threshold was defined as the lowest sound intensity required to generate a reproducible ABR waveform. Latencies of ABR waves Ⅰ and Ⅱ were measured at 70 dB click and 70 dB, 16 kHz pure-tone stimulus, respectively. Amplitudes were calculated as the peak-to-peak amplitude from the P1 or P2 peak to the next negative trough.

The DPOAE at 2f1-f2 was elicited from test mice using BioSig-RP software and a TDT system (Tucker-Davis Technologies). Five frequency points of 4, 8, 16, 28, and 32 kHz were selected to measure the 2f1-f2(f2/f1 = 1.2) to predict the auditory thresholds. Hearing thresholds were defined as the averaged signal for each identified frequency tested and compared with the corresponding frequency in the controls.

### Cell culture

SGNs were isolated from the mouse's inner ears following a detailed procedure outlined in a previous study (*Lv et al., 2010*). Mice at various ages were humanely sacrificed, and the temporal bones were removed. The dissecting solution contained minimum essential medium (MEM) with Hank's Balanced Salt Solution (HBSS) (Invitrogen) supplemented with 0.2 g/L kynurenic acid, 10 mM $MgCl_2$, 2% fetal bovine serum (FBS; v/v), and glucose (6 g/L). The SGN tissue was dissected and split into two segments: the apex and the base across the modiolar axis. The two-segment tissue was digested separately in an enzyme mixture containing collagenase type I (1 mg/mL) and DNase (1 mg/mL) at 37°C for 15 min. After gentle trituration and centrifugation (2000 rpm for 5 min) in 0.45 M sucrose, the cell pellets were reconstituted in 900 µL of culture media (Neurobasal A, supplemented with 2% B27 [v/v], 0.5 mM L-glutamine, 100 units/mL penicillin; Invitrogen). The suspension containing the SGNs

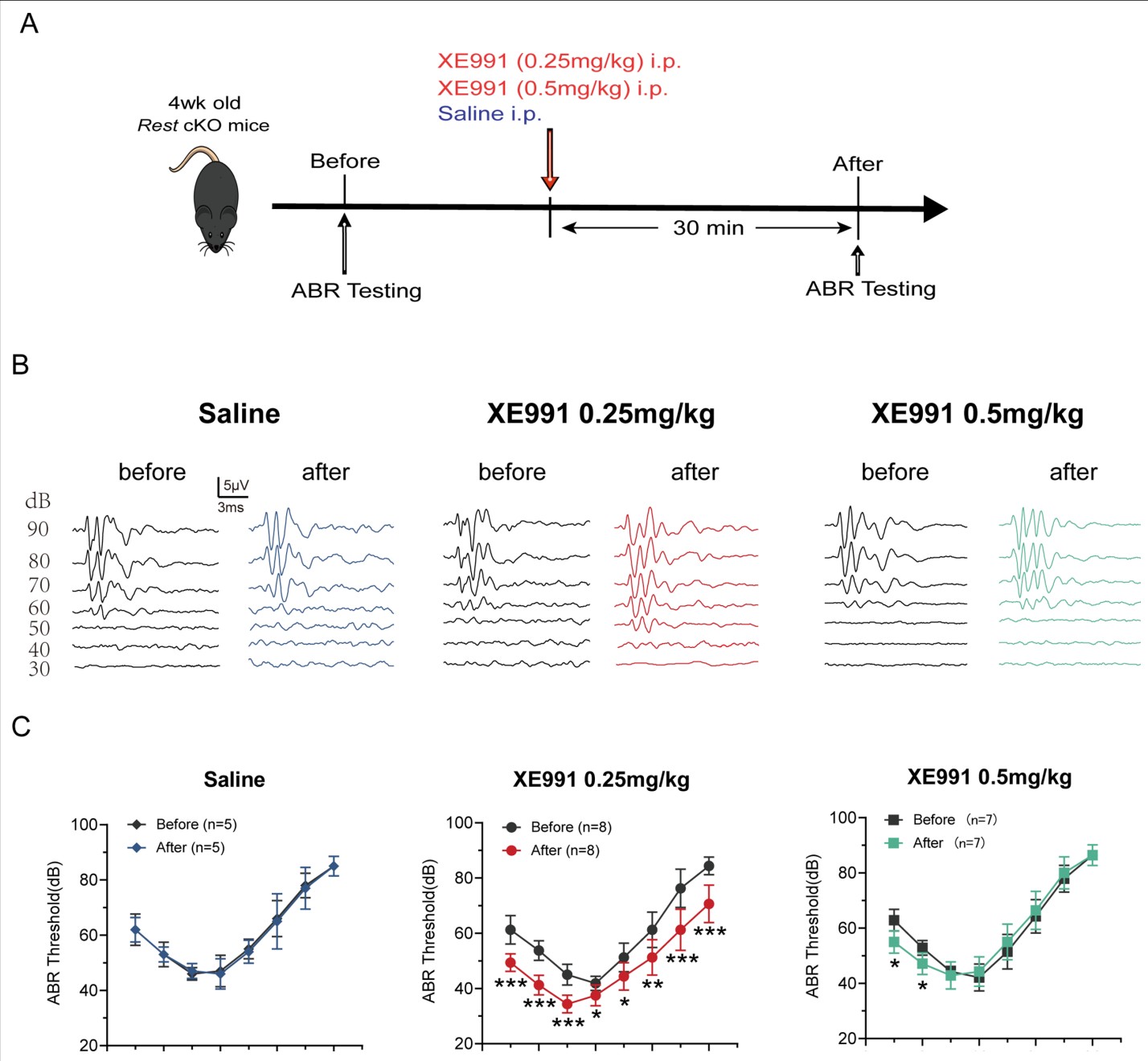

**Figure 8.** K$_v$7 channel blocking rescues hearing of *Rest* conditional knockout (cKO) mice. (**A**) Timeline for auditory brainstem response (ABR) testing and drug treatment. XE991 was administered to 1-month-old *Rest* cKO mice intraperitoneally at 0.25 mg/kg or 0.5 mg/kg. (**B**) Representative ABR waveforms in response to clicking (90–30 dB) sound pressure levels in the saline and XE991-treated groups mice. (**C**) ABR thresholds in the saline and XE991 (0.25, 0.5 mg/kg) treatment groups. Data are means ± SEM, *p<0.05, **p<0.01, ***p<0.001.

The online version of this article includes the following source data and figure supplement(s) for figure 8:

**Source data 1.** ABR data in control and XE991-treated mice.

**Figure supplement 1.** Amplitudes and latencies of auditory brainstem response (ABR) waves Ⅰ and Ⅱ in saline and XE991-treated mice.

**Figure supplement 1—source data 1.** Data of latencies and amplitudes of ABR waves in saline and XE991-treated mice.

was filtered through a 40 µm cell filter and planted on glass coverslips pretreated with 0.5 mg/mL poly D-lysine (Sigma-Aldrich) and 1 mg/mL laminin (Sigma-Aldrich). SGNs were kept in culture for 24–48 hr before electrophysiological recordings to allow for the detachment of Schwann cells from the soma.

A Chinese hamster ovary (CHO) cell line stably expressing human $K_v7.4$ (GenBank accession numbers: NM_004700.4) channels (Inovogen Tech, China) was maintained in MEM with 10% FBS, a 1% penicillin-streptomycin mixture (Gibco), and 20 µg/mL puromycin at 37°C with 5% $CO_2$. CHO cells were cultured in MEM supplemented with 10% FBS without antibiotics for 12–24 hr before transfection. The pIRES2-REST-EGFP (Sangon Biotech, Shanghai, China) or the pIRES2-EGFP (Sangon Biotech) plasmid constructs were transiently transfected into CHO cells alone or in combination, at a total amount of 600 ng/well, using Lipofectamine 3000 (Invitrogen) according to the manufacturer's instructions. Inovogen Tech performed authentication of purchased cell lines. All cell lines were certified mycoplasma-free as per our monthly contamination testing.

## Electrophysiology

Whole-cell voltage-clamp recordings from SGNs were performed at room temperature using a Multiclamp 700 B amplifier (Molecular Devices, Sunnyvale, CA, USA). Signals were filtered at 2 kHz with a low-pass Bessel filter and digitized at ≥20 kHz using a 12-bit acquisition system, Digidata 1332 (Axon Instruments), and pClamp 10.7 software (Molecular Devices). Patch pipettes were pulled from borosilicate glass capillaries and heat polished to a tip resistance of 3–4 MΩ. The pipette solution contained (in mM) 112 KCl, 2 $MgCl_2$, 0.1 $CaCl_2$, 10 HEPES, 1 EGTA, 5 $K_2$ATP, and 0.5 Na-GTP, adjusted to a pH of 7.35 with potassium hydroxide (KOH). The bath solution contained 130 mM NaCl, 5, 1 mM $MgCl_2$, 2 mM $CaCl_2$, 10 mM HEPES, and 10 mM glucose, adjusted to a pH of 7.4 with NaOH. To record the $K^+$ currents, SGNs were held at –20 mV, then hyperpolarized to –60 mV for 500 ms with a square voltage pulse before returning to holding potential. The deactivating $K^+$ current amplitude was calculated by subtracting the current amplitude measured at 10 ms before the end of the voltage step to –60 mV from the current amplitude measured at 10 ms after the onset of this voltage step. XE991 was purchased from Tocris Bioscience, and all other chemicals were purchased from Sigma.

Whole-cell current-clamp recordings from SGNs were conducted using the same pipette solution in the voltage-clamp experiment. To measure the $K_v7.4$ channel currents in CHO cells stably expressing $K_v$ 7.4 channels, pipettes were filled with internal solutions containing (in mM) 145 KCl, 1 $MgCl_2$, 10 HEPES, 5 EGTA, 5 $K_2$ATP, adjusted to a pH of 7.35 with KOH. The external solution contained 140 mM NaCl, 3 KCl, 1.5 $MgCl_2$, 2 $CaCl_2$, 10 HEPES, and 10 mM glucose and was adjusted to 7.4 pH with NaOH. $K_v7.4$ channel current traces were generated with 2.5 s depolarizing voltage steps from a holding potential of –80 mV to a maximal value of +40 mV in 10 mV increments. Currents were measured with capacitance and series resistance compensation (nominally 70–90%).

To record $K^+$ currents in OHCs, apical cochlear turns of P10–P14 mice were dissected in a solution containing (in mM) 144.6 NaCl, 5.5KCl, 1 $MgCl_2$, 0.1 $CaCl_2$, 0.5 $MgSO_4$, 10.2 HEPES, and 3.5 L-glutamine, adjusted to a pH of 7.2 with NaOH. The pipette solution contained (in mM) 142 KCl, 3.5 $MgCl_2$, 1 EGTA, 2.5 MgATP, 0.1 $CaCl_2$, 5 HEPES, adjusted to a pH of 7.4 with KOH. The external solution contained (in mM) 145 NaCl, 5.8 KCl, 0.9 $MgCl_2$, 1.3 $CaCl_2$, 0.7 $NaH_2PO_4$, 10 HEPES, and 5.6 D-glucose, adjusted to a pH of 7.4 with NaOH. To record $K^+$ currents in OHCs in a whole-cell configuration, OHCs were held at –84 mV with step voltages ranging from –144 mV to 34 mV, with 10 mV increments. Recordings were performed at room temperature using Axon patch 700 B (Molecular Equipment, USA) amplifiers. Data acquisition was controlled by Clampex10.7 (Molecular Equipment) and a Digidata 1440A A-D converter (Molecular Equipment).

## Whole-mount preparation of SGNs and HCs

The mice were anesthetized, and the cochleae were fixed in 4% paraformaldehyde (PFA) overnight at 4°C and then decalcified in 10% EDTA (#798681, Sigma, Darmstadt, Germany) in PBS for 2–3 days at 4°C. The EDTA solution was changed daily. Decalcified cochleae were processed using a sucrose gradient and embedded in an OCT compound (Tissue-Tek) for cryo-sectioning. The specimens were sliced into 10 µm sections for study.

For HC preparation, the SGNs, Reissner's membrane, and the most basal cochlear segments were removed after fixation with 4% PFA for 2 hr, followed by decalcification in 10% EDTA for 3 days at 4°C. HC preparations were used for immunofluorescence staining.

## Cell count

To evaluate SGN morphometry and density, cochlear sections were stained with hematoxylin and eosin (HE). Rosenthal's canal was divided into the apex, middle, and base regions. SGN density from these three regions was measured using Image-Pro Plus 5.1. Five to six sections per cochlea per animal were analyzed, and six mice were used in each group. Cells displaying Myo7A labeling were quantified as the number of positive cells per 100 μm of basilar membrane length from the apex, middle, and base for HC counts.

## Immunofluorescence staining

Specimens were simultaneously permeabilized and blocked with 10% goat serum in 0.1% TritonX-100 and 0.1% BSA for 30 min at 37°C and then labeled with primary antibodies overnight at 4°C. The primary antibodies used in this study were mouse anti-$K_v$7.4 antibodies (#ab84820, Abcam, Cambridge, UK; 1:100), rabbit anti-REST antibodies (#22242-1-AP, Proteintech, USA; 1:200), mouse anti-Tuj1 (#801202, BioLegend, CA; 1:100), rabbit anti-Myo7A antibodies (#25-6790, Proteus BioSciences, CA; 1:50), and Phalloidin-iFluor 555 (#ab176756, Abcam). For detection, the specimens were incubated with Alexa 488-conjugated and Alexa 568-conjugated secondary antibodies (#115-545-003, 115-165-003, 111-545-003, 111-165-003, Jackson ImmunoResearch, PA; 1:400) for 2 hr at room temperature and then stained with 4,6-diamidino-2-phenylindole dihydrochloride (DAPI, #D9542, Sigma) for 10 min at room temperature. After washing with PBS, ProLong Gold Antifade Mount (# P36934, Invitrogen, CA) was used to mount the samples. Sections were visualized using a confocal fluorescent microscope (Leica Microsystems, Wetzlar, Germany). Images were analyzed with Leica LAS AF Lite and processed using ImageJ and Photoshop CS5 (Adobe, San Jose, CA).

## Single-cell RT-PCR

Cochleae were dissected from adult WT mice. SGNs or HCs were aspirated into a patch pipette using a patch-clamp system under the microscope. The electrode tip was then quickly broken into an RNase-free PCR tube containing 1 μL of oligo dT primers (50 μM), 1 μL of dNTP mixture (10 mM), 2 μL of RNase-free water. The mixture was incubated in the water bath at 65°C for 5 min and then placed on ice for 1 min. Single-strand cDNA was synthesized from cellular mRNA by supplementing with 1 μL of PrimeScript II RTase (200 U/μL), 0.5 μL of RNase Inhibitor (40 U/μL), 2 μL of 5× PrimeScript II buffer, and 1.5 μL of RNase Free $H_2O$ and then incubating the mixture at 55°C for 50 min followed by 85°C for 5 min. The protocol was performed at 95°C for 5 min, followed by 35 cycles (95°C for 50 s, 59°C for 50 s, 72°C for 50 s) and final elongation at 72°C for 10 min by adding 'outer' primers (*Source data 1*). The PCR product was identified by 2% agarose gel electrophoresis. The reverse transcription kit was purchased from Takara-Clontech (6210A, Invitrogen), and the PCR kits were obtained from Promega (M7122, Madison).

## Western blotting

Cochlear tissues (five mice per sample) were homogenized, and cells were lysed in a RIPA solution containing 0.1% SDS, 50 mM Tris (pH 7.4), 1 mM EDTA (pH 8.0), 1% sodium deoxycholate, 1% TritonX-100, and 200 μM of phenylmethanesulfonyl fluoride (PMSF). The lysis mixture was centrifuged at 13,800 × *g* for 30 min at 4°C. The supernatant was removed, and the protein concentration was determined using a BCA Protein Assay Kit (#23235, Thermo, CA). Equal amounts of protein (40 μg) were resolved by SDS-PAGE and transferred to a PVDF membrane (#IPVH00010, Millipore, MA). Membranes were blocked with 1× TBST containing 5% dry milk for 2 hr at room temperature and probed with the following antibodies: anti-$K_v$7.4 antibodies (#ab84820, Abcam; 1:500), anti-REST antibodies (#ab21635, Abcam; 1:1000), or anti-tubulin beta-III antibodies (#MMS-435P, BioLegend; 1:10,000) overnight at 4°C. Then, fluorescent secondary conjugated IgG (#IRDye 800CW goat anti-rabbit IgG [H+L]; #IRDye 800CW goat anti-mouse IgG [H+L], LI-COR Biosciences, NE; 1:10,000) was applied for 90 min at room temperature. All Western blots were quantified using Image Studio software (IS, LI-COR Biosciences).

## Quantitative real-time PCR

Total RNA from mouse cochleae was extracted using an RNA Extraction Kit according to the manufacturer's instructions (#9767, TaKaRa, Japan). The extracted RNA concentration was measured using a

NanoDrop 2000 spectrophotometer (#ND-LITE, Thermo, DE), and 1000 ng of total RNA was reversed-transcribed using PrimeScript reverse transcriptase (#RR036Q, TaKaRa, China). The synthesized cDNA was used as a template to perform quantitative real-time (qRT)-PCR with TB Green Premix Ex Taq (#RR420Q, TaKaRa) in a Bio-Rad Real-Time PCR System (Bio-Rad Laboratories, CA). Glyceraldehyde 3-phosphate dehydrogenase (GAPDH) was used as an internal control to normalize the relative mRNA abundance of each cDNA. The relative mRNA expression levels were calculated using the standard formula $2^{-\Delta\Delta Ct}$. All the primers used in this study were designed using Primer Premier 6.25 (PREMIER Biosoft International, CA) and synthesized by Sangon Biotech (Shanghai, China). The primer sequences are shown in *Source data 1*.

### In vivo drug administration

For fasudil administration
6–8-week-old C57 mice were divided into four groups: control, saline, fasudil 10 mg/kg, and fasudil 20 mg/kg. Fasudil was obtained from the National Institute for the Control of Pharmaceutical and Biological Products (Beijing, China), dissolved in 0.9% saline, and administered intraperitoneal injection every 2 days for 28 days. Control mice were injected intraperitoneally with the same volume of saline. ABR was recorded before and after treatment with the drug and on days 14 and 28 after treatment to assess hearing function. The cochlea was also removed on day 28 to evaluate the morphological changes of the HCs and SGNs and the functional changes of the SGNs.

For XE991 administration
The 1-month-old *Rest* cKO mice were divided into three groups: the saline, XE991 0.25 mg/kg, and 0.5 mg/kg. XE991 dihydrochloride was obtained from Abcam and dissolved in 0.9% saline. Mice received a single dose of XE991 (0.25, 0.5 mg/kg) or saline intraperitoneally. ABR was measured in each group of mice before and 30 min after drug administration to characterize the effects of the drug on mouse hearing.

### Statistical analysis

All data are presented as the means ± SEM. Statistical analyses were performed using GraphPad Prism 6.0. The Student's *t*-test was used to compare data between two groups, and ANOVA was used for multiple comparisons. Proportions of cells displaying distinct electrophysiological properties were analyzed using Fisher's exact test or Pearson's chi-square test. *, **, and *** indicate statistically significant results compared to the appropriate controls and indicate $p < 0.05$, $p < 0.01$, and $p < 0.001$, respectively.

## Acknowledgements

This work was supported by the National Natural Science Foundation of China (81670939) to PL, the Natural Science Foundation of Hebei Province of China (H2021206286) to PL, the Department of human resources and social security of Hebei Province for Talents (A202005003) to PL, the central government guiding local funding projects for scientific and technological development(216Z7701G) to PL, and the Science Fund for Creative Research Groups of Natural Science Foundation of Hebei Province (H2020206474) to PL. ENY was supported by grants from the National Institutes of Health (P01 AG051443, R01 DC015135, R01 DC016099, and R01 AG060504-01). NG was supported by the BBSRC International Partnering Award BB/R02104X/1 and the 100 Foreign Experts of Hebei Province program.

## Additional information

### Funding

| Funder | Grant reference number | Author |
| --- | --- | --- |
| National Institute on Deafness and Other Communication Disorders | DC015135 | Ebenezer N Yamoah |
| National Institute on Aging | AG060504-01 | Ebenezer N Yamoah |
| National Institute on Deafness and Other Communication Disorders | DC016099 | Ebenezer N Yamoah |
| National Institute on Aging | P01 AG051443 | Ebenezer N Yamoah |

The funders had no role in study design, data collection and interpretation, or the decision to submit the work for publication.

### Author contributions

Haiwei Zhang, Data curation, Formal analysis, Methodology, Writing - original draft; Hongchen Li, Mingshun Lu, Shengnan Wang, Xueya Ma, Fei Wang, Jiaxi Liu, Xinyu Li, Data curation, Formal analysis; Haichao Yang, Data curation, Formal analysis, Methodology; Fan Zhang, Formal analysis, Funding acquisition, Investigation, Methodology; Haitao Shen, Conceptualization, Data curation, Funding acquisition, Investigation; Noel J Buckley, Resources, Funding acquisition, Investigation, Writing - original draft, Project administration; Nikita Gamper, Resources, Supervision, Funding acquisition, Investigation, Writing - original draft, Project administration, Writing - review and editing; Ebenezer N Yamoah, Conceptualization, Data curation, Formal analysis, Funding acquisition, Writing - original draft, Project administration, Writing - review and editing; Ping Lv, Conceptualization, Data curation, Formal analysis, Funding acquisition, Validation, Methodology, Writing - original draft, Writing - review and editing

### Author ORCIDs

Haiwei Zhang http://orcid.org/0000-0002-8209-9395
Hongchen Li http://orcid.org/0000-0003-2639-8726
Nikita Gamper http://orcid.org/0000-0001-5806-0207
Ebenezer N Yamoah http://orcid.org/0000-0002-9797-085X
Ping Lv http://orcid.org/0000-0002-6271-8865

### Ethics

All experimental animal protocols were performed following the Animal Care and Ethical Committee of Hebei Medical University (Shijiazhuang, China). 01644.

### Decision letter and Author response

Decision letter https://doi.org/10.7554/eLife.76754.sa1
Author response https://doi.org/10.7554/eLife.76754.sa2

## Additional files

### Supplementary files

• Transparent reporting form
• Source data 1. Primer information for genotyping of the mouse tails and single-cell RT-PCR and quantitative real-time PCR.

### Data availability

All data generated or analyzed are included in the source data.

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
