## [Editor Report]

Genetic forms of deafness are a major health challenge. This study deciphers the cochlear roles of repressor element 1–silencing transcription factor (REST), a gene involved in the DFNA27 dominant form of deafness, using the mouse as a model system. This study provides evidence for a pathophysiological mechanism of deafness and shows how genes involved in different forms of deafness may interact together. The article will be interesting to readers who work in the field of hearing research, REST regulation, or K_v_7.4 regulation.

---

## [Decision Letter]

**Decision letter after peer review:**

Thank you for submitting your article "Repressor element 1–silencing transcription factor deficiency yields profound hearing loss through K_v_7.4 channel upsurge in auditory neurons and hair cells" for consideration by *eLife*. Your article has been reviewed by 3 peer reviewers, and the evaluation has been overseen by Tanya Whitfield as the Reviewing Editor and Kathryn Cheah as the Senior Editor. The following individuals involved in the review of your submission have agreed to reveal their identity: Radha Kalluri (Reviewer #2); Nicolas Antoine Michalski (Reviewer #3).

The reviewers have discussed their reviews with one another, and agree that although the work is promising, some revisions (including additional experimental work and important controls) are required to support the conclusions drawn. The Reviewing Editor has drafted this to help you prepare a revised submission.

Essential revisions:

1. Demonstrate appropriate controls:

– For REST expression in SGNs and hair cells, together with details of the epitope recognised by the antibody

– To ensure RT–PCR amplification products represent those from mRNA rather than genomic DNA, addressing the issues of alternative splicing

2. Address the issue of the expression of Tg(Atoh1–Cre) in supporting cells and SGNs, and the consequences this may have for the REST cKO phenotype.

3. Acknowledge the known functions, specificity and duration of action of fasudil when the compound is first mentioned, discussing any limitations this may have for interpreting the effects of this compound and citing the relevant literature.

4. Provide further analysis and discussion of the effects of REST cKO in addition to the up–regulation of Kv7 currents, clarifying whether (or not) there is any causal link between the hearing loss originating from an alteration of the bioelectrical properties of the sensory cells and the hearing loss originating from cellular deterioration.

5. Perform a more comprehensive analysis of the ABR responses to characterise the hearing status of the various mouse models (REST cKO, XE199–treated and fasudil–treated) at both 1 month and 3 months of age, addressing any distinction between OHC and SGN deficits.

6. Address the list of minor suggestions from the Reviewers, including various corrections concerning the citation of the literature.

The full reviews are appended below and provide more information.

*Reviewer #1 (Recommendations for the authors):*

Detailed comments:

Related to Figure 1: A number of studies have shown that REST is expressed in nearly all non–neuronal cells where it represses hundreds of neuronally–transcribed genes. In differentiating neurons and hair cells (HCs), REST activity is downregulated. In most neurons, transcriptional repression of Rest is the primary mechanism underlying the downregulation of REST activity. In HCs, regulated splicing of an alternative exon into the Rest mRNA is the main mechanism by which REST activity in suppressed. Based on these, one would expect to find minimal expression of Rest in SGNs; however, Zhang and colleagues show strong REST immunostaining in SGNs (Figure 1A). HCs were not tested for REST protein expression by the authors, but they used single cell RT–PCR to evaluate Rest mRNA expression, and they detected Rest in both HCs and SGNs (Figure 1B). These data should be supported by robust controls. For example, the Rest cKO mice could be used to test the specificity of REST immunostaining in SGNs. A similar experimental design is recommended for testing the expression of REST in HCs. The location of the epitope of the anti–REST antibody within the target protein is also an important (and not provided) piece of information because alternative splicing is predicted to prevent the expression of the C–terminal half of REST (encoded by exon 5) but not the N–terminal half. Notably, Rest contains 5 exons, not 4 exons; the latter number is stated erroneously in the manuscript at multiple places.

Figure 1 related: The authors' single cell RT–PCR data would be more informative if the Rest primers were designed from 2 different exons and not from the same exon (all of the Rest RT–PCR primers are designed from exon 5 in this study). If the primers were designed from different exons, amplification of the genomic DNA could be ruled out. In Figure 1B and Methods, there is no control included to test whether the RT–PCR products are amplicons of mRNA or genomic DNA. The authors' primers are also not suitable to test the splicing of Rest. Using these primers, the generated RT–PCR products will not differ in size regardless of whether the alternative exon of Rest (i.e., exon 4) is spliced into the mRNA or not. Therefore, the generated amplicons cannot indicate whether the expressed mRNA encodes a functional REST protein or a non–functional REST that is "truncated" by the previously described alternative splicing of exon 4 of Rest.

Related to Figure 1–9: Tg(Atoh1–Cre) expression is not completely specific to HCs within the organ of Corti. This is not a problem if the effect of Tg(Atoh1–Cre)–dependent gene deletion is robust (e.g., widespread loss of HCs). In contrast, when the HC loss is mild (like in this study), a significant concern is that Tg(Atoh1–Cre) is expressed in some % of supporting cells (see Figures 6 and 7 in PMID: 16145671). Data in this manuscript do not rule out the possibility that much of the HC loss and hearing loss of Rest cKO mice are caused by the deletion of Rest in some % of supporting cells in the organ of Corti.

Related to Figure 1–9: What % of SGNs express Tg(Atoh1–Cre)?

Related to Figure 1: The authors state that "both heterozygous and homozygous Rest knockout mice showed a significant increase in the ABR threshold (as compared to WT)". The authors analyzed the hearing loss of Rest +/cKO (i.e., heterozygous) mice only at a very early time (i.e., 1–2 months). In this age range, the reported hearing loss is mild and not entirely convincing. The demonstration of hearing loss of Rest +/cKO mice would be an important piece of new information because these mice are potentially useful to model the disorders of human subjects who are heterozygous for REST–inactivating mutations. Therefore, it would be valuable to determine the hearing thresholds (click and pure tone) of Rest +/cKO mice at precise timepoints (at 3 and 4 months) when hearing loss is perhaps more robust than in the 1–2 months age range.

Related to Figure 7: The authors suggest that the Kv7.4 activator effect of fasudil is contrary to its Rho kinase (ROCK) inhibitory activity: "Fasudil was initially described as a Rho–associated protein kinase (ROCK) inhibitor. However, recently Li and colleagues found that fasudil selectively increased Kv7.4 channel currents…" (Discussion section). It is unclear why the authors contrast these two effects of fasudil; both effects are supported by multiple studies: fasudil inhibits ROCK and activates Kv7.4. It seems to be important to state both effects of fasudil at the first mention of this compound in the text.

Related to Figure 7: A previous study reported that the Kv7.4–activating effect of fasudil rapidly diminishes after the washout of this drug (PMID: 27677924). Therefore, it would be important that the authors clarify/discuss the mechanism by which in–vivo administered fasudil could affect K^+^ currents in SGNs that were cultured for 24–48 h after isolation from fasudil–injected mice (Figure 7C–I).

Figure 5—figure supplement 3 shows that overexpressed REST suppresses the expression of plasmid–encoded Kcnq4 (i.e., Kv7.4) in CHO cells. The Kcnq4–encoding plasmid is not described in the manuscript, and it is unclear whether it contains REST binding sites.

Related to Figure 8 and 9: The strongest presented evidence for the hearing loss–causing effect of abnormally high Kv7.4 expression is the XE991–dependent (partial) rescue of hearing in Rest cKO mice. This is a nice experiment but it does not reveal whether XE991 acts on the cells of interest (HCs and SGNs) and on the protein of interest (i.e., Kv7.4) to achieve the improvement in hearing. Ideally, a genetic approach should be used to support the Kv7.4–related conclusion of the study.

*Reviewer #2 (Recommendations for the authors):*

Progressive hearing loss and cellular deterioration: As indicated above, I found the argument that progressive loss and deterioration of the sort seen in 3 month REST kno as being due to loss of Kv7 to be less convincing. I would have found this second point to be more convincing if the experiments with Kv7 current enhancer fasudil had also shown cellular deterioration like that in the REST knockouts. The data presented in the supplementary figures show that this is not the case, at least not over the timeline of the present experiments. Perhaps the deterioration would take longer than the 28 days over which fasudil was administered but no data was presented to argue this point. Alternatively, knocking out REST may do more than enhance Kv7 currents. More careful handling of this issue is needed.

I also found the term 'hearing loss' in reference to the elevated ABR and DPOAE thresholds to be somewhat unsatisfying. The term does not capture the full impact of enhancing Kv7 channels. Indeed, the ABR responses in the REST knockout (1 month) or fasudil enhanced Kv7 model are robust, with large peaks and multiple waves clearly visible, albeit with elevated thresholds. I wonder, for example, if the desensitized cochlea (with elevated thresholds) has a compressed dynamic range, greater synchrony, and shorter latencies? Similarly, in the case with Kv7 channels blocked, what are the broad response features of the 'sensitized' cochlea. I found this aspect of the data to be interesting but unaddressed. I would have liked to see a more complete analysis of the ABR waves amplitude and latency as a function of stimulus intensity to determine if there is also a difference in the level dependence that would better indicate the hearing status of these mouse models.

Overall, the experiments are nicely done and the data is presented clearly. However, I found the use of extensive figure supplements with minimal description to be confusing. The going back and forth between the main and supplement figures took away from the otherwise clear presentation of data. The authors should consider integrating the necessary figure supplements into the main figures and reducing the number of supplementary figures to those that are essential.

*Reviewer #3 (Recommendations for the authors):*

Recommendations to be addressed:

– The whole paragraph 125–137 is very confusing in the wording between knock–out mice and conditional knock–out mice. The paragraph should clearly state that:

i) First ABRs were recorded in knock–out mice…

ii) We then recorded in ABRs in conditional knock–out mice…

For instance, the use of "further" in line131 is misleading because it gives the impression we are dealing with the same mice whereas it is not the case. Similarly, what are Rest–homozygous mice? I suppose they are Rest knock–out mice?

– While reading the manuscript, I thought the authors could better insist on the OHC deficits and their relationship to the hearing loss observed. The authors insist very much on the SGN excitability deficits. However, these deficits are already present in 1 month old mice with a similar severity to 3 month old mutant mice although ABR thresholds are still normal at 1 month of age. A likely explanation is rather the DPOAE deficit at three months of age that becomes very strong compared to one month. To better disentangle these phenomena, the authors should analyze the amplitude of wave–1and its timing in Rest cKO mice based on their already acquired data. If these parameters are normal at one month of age, it suggests that the most dramatic phenotype relies on OHCs. Alternatively, wave–1 amplitude could be altered (with normal auditory threshold). In that case, it would show that both the OHC and SGN deficits cumulatively contribute to the hearing impairment. In addition, Kcnq4 playing on the excitability of SGNs, one would expect the timing of wave–1 to be altered.

Depending on these analysis, the authors should better discuss the relative contribution of SGN and OHCs to the auditory phenotype.

– Line373: The statement "the exact site of action of systemic fasudil on hearing requires further elucidation" is not satisfactory. As mentioned earlier, by analyzing wave 1 amplitude and timing in their signal, the authors could specifically address whether fasudil affects HCs and SGNs. Expression of Kv7.4 in the dorsal cochlear nucleus is not expected to affect the ABR signal before Wave–II. Please better discuss this point after analyzing these parameters in the datasets.

[Editors' note: further revisions were suggested prior to acceptance, as described below.]

Thank you for resubmitting your work entitled "Repressor element 1–silencing transcription factor deficiency yields profound hearing loss through K_v_7.4 channel upsurge in auditory neurons and hair cells" for further consideration by *eLife*. Your revised article has been evaluated by Kathryn Cheah (Senior Editor) and a Reviewing Editor.

The reviewers agree that the manuscript has been improved and will make a valuable contribution to the field. However, there are several remaining issues that need to be addressed, concerning both presentation and interpretation of the data, as outlined below. Please address these concerns, and make sure that any figures used in the rebuttal have also been added to the manuscript, as requested by Reviewer 1. Reviewer 2 also suggests that you may wish to consider moving some of the supplementary material into the main figures. Reviewer 3 also lists a number of typos to be corrected.

*Reviewer #1 (Recommendations for the authors):*

The authors of this manuscript conclude that genetic inactivation of Rest in hair cells (HCs) and spiral ganglion neurons (SGNs) cause progressive hearing loss by upregulating the expression of Kv7.4 in the two cell types ("Repressor element 1–silencing transcription factor deficiency yields profound hearing loss through Kv7.4 channel upsurge in auditory neurons and hair cells"). This conclusion is not fully supported by the presented data because the utilized Cre–expressing transgene and some of the key activators/inhibitors in the study are not sufficiently selective to draw a definitive conclusion: (1) The Atoh1–Cre transgene, which was used for the generation of Rest cKO mice in the study, is expressed not only in HCs and SGNs but also in many non–HCs and non–SGNs in the cochlea. This transgene is an excellent tool for many purposes but it is not ideal to determine the effects of SGN/HC specific deletion of Rest on hearing because Rest is expressed in non–HCs and non–SGNs in the hearing organ. (2) Fasudil, which was used for the activation of Kv7.4 in the study, is not a Kv7.4 specific drug; it also binds to (and inhibits) Rho–associated kinases. Thus, the hearing loss of fasudil–injected mice may not be caused by the activation of Kv7.4 but by the inhibition of Rho–associated kinases. (3) XE991 is not a specific inhibitor of Kv7.4, it also inhibits other Kv7 channels. Therefore, it is unclear whether the 10–15 dB improvement in the hearing threshold of XE991–treated Rest cKO mice is caused by the inhibition of Kv7.4 or other Kv7 channels. In summary, the authors' overall conclusion is not entirely convincing because much of their data can be interpreted differently than specified in the manuscript.

In the authors' rebuttal letter (and in the manuscript), one unvalidated data interpretation is used to support another unvalidated interpretation; e.g., in the 3rd answer to reviewer–1, the authors view upregulation of Kv7.4 as a proven cause (as opposed to a 'possible cause') of hearing impairment in Rest cKO mice and use this to state that the hearing loss–relevant deletion of Rest must have been in HCs and/or SGNs because other cochlear cells do not express Kv7.4. This kind of reasoning goes awry easily.

The authors show the ABR thresholds (click and pure tone) of 3–4 months old Rest Het mice in the rebuttal letter but not in the actual manuscript. It would be more useful to the readers if these figure panels were added to the manuscript.

*Reviewer #2 (Recommendations for the authors):*

1. One of the most interesting aspects of this data set was the patch–clamping data showing enhanced expression of Kv7 currents in Rest cKO mice and the accompanying changes in ABR thresholds at 1M of age, before the onset of obvious cell deterioration. This combined with the experiments showing partial recovery of thresholds using the Kv7 blocker xe991 was the strongest indication that modulation of this particular ion channel was shaping hearing sensitivity, even in the absence of cell death. I had urged the authors to examine the stimulus level–dependent changes in ABR amplitudes and latencies to more fully characterize the hearing status under these different manipulations, hoping that this would help disentangle the impact of cell deterioration versus enhancement of Kv7 currents on the hearing loss phenotype. The authors have partially satisfied that request by providing more ABR data in supplements. However, by providing this analysis at a fixed level and with little further discussion, the authors missed the point…which was to test if there were differences in how the amplitude and latencies grow with stimulus intensity which might have indicated interesting differences in ABR morphology that could be attributed to the increase/decrease in phasic firing in Rest KO or XE199 manipulations. As currently presented in the text, the new ABR data does little to interpret the finding further.

2. Is elevation in Kv7 currents neuroprotective or does this lead to cellular deterioration? Lines 297–306 in the discussion left me with the impression that the authors are arguing that overexpression of Kv7 currents can cause a similar degeneration in cochlear cells as previously shown due to loss of function mutations in Kv7.4. This is perhaps not what the authors intended, because they agreed in their response to my original review that knocking out REST may cause cellular deterioration from multiple pathways and that the fasudil manipulation did not show cellular deterioration. Thus, these studies cannot determine if the cellular degeneration noted in the 3–4 month Rest KO mouse is due to overexpression of Kv7…so the question of whether Kv7 may be neuroprotective cannot be addressed here. I'd recommend a careful revision of this paragraph to avoid conflating elevation in threshold that they may attribute to the enhancement of Kv7 from that which may result from overall cellular damage.

3. The reduced recovery of thresholds for the higher dose of Xe991 is counterintuitive. Why doesn't the higher concentration of XE991 restore thresholds and only at lower frequencies as did the lower concentration? Rather than addressing what seems counterintuitive to me, the authors assert that there is an improvement in threshold and gloss over what seems to be only a partial rescue at the higher dose. Some sort of discussion of this is needed…for example is there some balance between synchrony and excitability that may produce this result? It will be important to do this since the rescue with the blocker is one of the most convincing pieces of data in this study.

4. Problem with the summary figure shown in figure 9. The schematic shows an OHC providing direct input to Type I auditory neurons. Since the input from OHCs to most of the auditory nerve is indirectly coupled through fluid mechanics, this figure could be misconstrued as meaning otherwise. Please revise to avoid such confusion. Alternatively, remove the figure.

5. I still found several supplementary figures to be disruptive to the presentation of the material. Lines 178:186; Figure 3 supplement and Figure 5 supplement 1. I don't see the rationale for discussing the P14 data separately from the 1 Month and 3 Months time points. It seems that it would be natural to integrate all three time points into the same figure. Same comment for Figure 5, why not fold in the P14 data in rather than show a nearly identical representation of that one–time point as a supplement?

---

## [Author Response]

Essential revisions:Reviewer #1 (Recommendations for the authors):Detailed comments:Related to Figure 1: A number of studies have shown that REST is expressed in nearly all non–neuronal cells where it represses hundreds of neuronally–transcribed genes. In differentiating neurons and hair cells (HCs), REST activity is downregulated. In most neurons, transcriptional repression of Rest is the primary mechanism underlying the downregulation of REST activity. In HCs, regulated splicing of an alternative exon into the Rest mRNA is the main mechanism by which REST activity in suppressed. Based on these, one would expect to find minimal expression of Rest in SGNs; however, Zhang and colleagues show strong REST immunostaining in SGNs (Figure 1A). HCs were not tested for REST protein expression by the authors, but they used single cell RT–PCR to evaluate Rest mRNA expression, and they detected Rest in both HCs and SGNs (Figure 1B). These data should be supported by robust controls. For example, the Rest cKO mice could be used to test the specificity of REST immunostaining in SGNs. A similar experimental design is recommended for testing the expression of REST in HCs. The location of the epitope of the anti–REST antibody within the target protein is also an important (and not provided) piece of information because alternative splicing is predicted to prevent the expression of the C–terminal half of REST (encoded by exon 5) but not the N–terminal half. Notably, Rest contains 5 exons, not 4 exons; the latter number is stated erroneously in the manuscript at multiple places.

Thank you for your suggestion. We tested the REST protein expression in SGN by immunostaining in WT and *Rest* cKO mice. Our results showed that *Rest* was abundantly expressed in SGN of WT mice, while *Rest* expression was not detected in *Rest* cKO mice. (Figure 1A) The anti-*Rest* antibody (proteintech, 22242-1-AP) recognized the C-terminal of REST. We have corrected the exons number of *Rest*. We thank the reviewer for pointing it out (please see Figure 1C, line 299). Additionally, we provide additional single-cell RT-PCR analyses of *Rest* in OHCs, and IHCs of WT and *Rest* cKO mice. The primer sequences: *Rest* forward: GGT CTG ATC CCG CTC CG, reverse：TGG CCA TAA CTG TAC TCC TCT G (Figure 1—figure supplement 1).

Figure 1 related: The authors' single cell RT–PCR data would be more informative if the Rest primers were designed from 2 different exons and not from the same exon (all of the Rest RT–PCR primers are designed from exon 5 in this study). If the primers were designed from different exons, amplification of the genomic DNA could be ruled out. In Figure 1B and Methods, there is no control included to test whether the RT–PCR products are amplicons of mRNA or genomic DNA. The authors' primers are also not suitable to test the splicing of Rest. Using these primers, the generated RT–PCR products will not differ in size regardless of whether the alternative exon of Rest (i.e., exon 4) is spliced into the mRNA or not. Therefore, the generated amplicons cannot indicate whether the expressed mRNA encodes a functional REST protein or a non–functional REST that is "truncated" by the previously described alternative splicing of exon 4 of Rest.

We agree with your comments. Our primers read from a single exon, and the amplification from the genomic DNA cannot be ruled out. We changed our primers sequence to that used in Nakano's paper (Nakano, Kelly, et al., 2018), designed from exon 3 to exon 5 of *Res*t. The sequence of the primers is forward 5' CGACACATGCGGACTCATTC 3', and reverse 5' AGAGGCCACATAATTGCACTG 3'. Our results show that two bands were detected in hair cells and SGNs, representing REST and its splice form, Rest^4^, respectively (Figure 1B). The OHCs were isolated from WT mice at P13, IHCs from mice at P20, and SGNs from mice at P40.

Related to Figure 1–9: Tg(Atoh1–Cre) expression is not completely specific to HCs within the organ of Corti. This is not a problem if the effect of Tg(Atoh1–Cre)–dependent gene deletion is robust (e.g., widespread loss of HCs). In contrast, when the HC loss is mild (like in this study), a significant concern is that Tg(Atoh1–Cre) is expressed in some % of supporting cells (see Figures 6 and 7 in PMID: 16145671). Data in this manuscript do not rule out the possibility that much of the HC loss and hearing loss of Rest cKO mice are caused by the deletion of Rest in some % of supporting cells in the organ of Corti.

As shown in the paper (PMID: 16145671), Tg(Atoh1-Cre) is expressed in some supporting cells. Our study indicates that hearing loss caused by *Rest* deletion is through upregulation of the K_v_7.4 channel. Specific to the reviewer's concern, there is no evidence that Kv7.4 channels are expressed in supporting cells (PMID: 32285007). Therefore, our study suggests that deafness in *Rest* cKO mice is mainly due to SGNs and HCs dysfunction.

Related to Figure 1–9: What % of SGNs express Tg(Atoh1–Cre)?

The reviewer has posed an important question, but we do not believe the questions addressed in this report can shed light on it.

Related to Figure 1: The authors state that "both heterozygous and homozygous Rest knockout mice showed a significant increase in the ABR threshold (as compared to WT)". The authors analyzed the hearing loss of Rest +/cKO (i.e., heterozygous) mice only at a very early time (i.e., 1–2 months). In this age range, the reported hearing loss is mild and not entirely convincing. The demonstration of hearing loss of Rest +/cKO mice would be an important piece of new information because these mice are potentially useful to model the disorders of human subjects who are heterozygous for REST–inactivating mutations. Therefore, it would be valuable to determine the hearing thresholds (click and pure tone) of Rest +/cKO mice at precise timepoints (at 3 and 4 months) when hearing loss is perhaps more robust than in the 1–2 months age range.

We measured the hearing thresholds of *Rest* +/- mice at 3 and 4 months. The data showed that the hearing loss was more robust in *Rest* +/- mice at 3-4 months of age than in mice at 1-2 months.

**Author response image 1. sa2fig1:** 

Related to Figure 7: The authors suggest that the Kv7.4 activator effect of fasudil is contrary to its Rho kinase (ROCK) inhibitory activity: "Fasudil was initially described as a Rho–associated protein kinase (ROCK) inhibitor. However, recently Li and colleagues found that fasudil selectively increased Kv7.4 channel currents…" (Discussion section). It is unclear why the authors contrast these two effects of fasudil; both effects are supported by multiple studies: fasudil inhibits ROCK and activates Kv7.4. It seems to be important to state both effects of fasudil at the first mention of this compound in the text.

We thank the reviewer for the comment, and we rewrote it as “Fasudil was initially described as a Rho-associated protein kinase (ROCK) inhibitor. Recently Li and colleagues found that fasudil selectively increased Kv7.4 channel currents…” (line 366)

Related to Figure 7: A previous study reported that the Kv7.4–activating effect of fasudil rapidly diminishes after the washout of this drug (PMID: 27677924). Therefore, it would be important that the authors clarify/discuss the mechanism by which in–vivo administered fasudil could affect K^+^ currents in SGNs that were cultured for 24–48 h after isolation from fasudil–injected mice (Figure 7C–I).

In this paper (PMID: 27677924), fasudil acts on cells with acute perfusion, and its activation of Kv7.4 channels is through interaction with Ile^308^ in its S6. Moreover, the acute effect of fasudil can be washed out. Besides the acute effects of fasudil as outlined in the report, we surmise that the drug may also have a chronic effect. In the current study, fasudil was injected into mice for 1 month, and the chronic effect of fasudil appeared different from the acute one. We speculate that the effect of fasudil on potassium currents in SGN may be through affecting the translocation or expression of Kv7 channels. We have discussed the subject (lines 381-384).

Figure 5—figure supplement 3 shows that overexpressed REST suppresses the expression of plasmid–encoded Kcnq4 (i.e., Kv7.4) in CHO cells. The Kcnq4–encoding plasmid is not described in the manuscript, and it is unclear whether it contains REST binding sites.

The Kcnq4-encoding plasmid is human-derived (Genebank accession numbers: NM_004700.4). We have included this information in the "method section"(lines 441-442). We also identified that KCNQ4 contains REST binding sites through the JASPAR website (Author response image 2 shows the data from JASPAR).

Related to Figure 8 and 9: The strongest presented evidence for the hearing loss–causing effect of abnormally high Kv7.4 expression is the XE991–dependent (partial) rescue of hearing in Rest cKO mice. This is a nice experiment but it does not reveal whether XE991 acts on the cells of interest (HCs and SGNs) and on the protein of interest (i.e., Kv7.4) to achieve the improvement in hearing. Ideally, a genetic approach should be used to support the Kv7.4–related conclusion of the study.

We thank the reviewer for the suggestions, and we agree that the evidence is pharmacological and a genetic strategy would have solidified our assertions. The genetic approach may be beyond the scope of the current studies.

Reviewer #2 (Recommendations for the authors):Progressive hearing loss and cellular deterioration: As indicated above, I found the argument that progressive loss and deterioration of the sort seen in 3 month REST kno as being due to loss of Kv7 to be less convincing. I would have found this second point to be more convincing if the experiments with Kv7 current enhancer fasudil had also shown cellular deterioration like that in the REST knockouts. The data presented in the supplementary figures show that this is not the case, at least not over the timeline of the present experiments. Perhaps the deterioration would take longer than the 28 days over which fasudil was administered but no data was presented to argue this point. Alternatively, knocking out REST may do more than enhance Kv7 currents. More careful handling of this issue is needed.

We agreed with your comments. Since we only administered fasudil for 28 days, we could not determine if the extended administration time caused cellular deterioration. In addition, because REST has many target genes, knocking out *Rest* may cause cellular deterioration in 3-month-old *Rest* cKO mice through other pathways.

I also found the term 'hearing loss' in reference to the elevated ABR and DPOAE thresholds to be somewhat unsatisfying. The term does not capture the full impact of enhancing Kv7 channels. Indeed, the ABR responses in the REST knockout (1 month) or fasudil enhanced Kv7 model are robust, with large peaks and multiple waves clearly visible, albeit with elevated thresholds. I wonder, for example, if the desensitized cochlea (with elevated thresholds) has a compressed dynamic range, greater synchrony, and shorter latencies? Similarly, in the case with Kv7 channels blocked, what are the broad response features of the 'sensitized' cochlea. I found this aspect of the data to be interesting but unaddressed. I would have liked to see a more complete analysis of the ABR waves amplitude and latency as a function of stimulus intensity to determine if there is also a difference in the level dependence that would better indicate the hearing status of these mouse models.

Thank you for your comments. We have reanalyzed the amplitude and latency of ABR waves shown in Figure 1—figure supplement 2, Figure 7—figure supplement 2, and Figure 8—figure supplement 1.

We demonstrate from the ABR waveform amplitude and latency assessment that the decline in hearing status ensues from early sound processing at the levels of hair cells and SGNs. The impact of XE991 in ABR waveforms, showing the modest extent of recovery of the hearing status

Overall, the experiments are nicely done and the data is presented clearly. However, I found the use of extensive figure supplements with minimal description to be confusing. The going back and forth between the main and supplement figures took away from the otherwise clear presentation of data. The authors should consider integrating the necessary figure supplements into the main figures and reducing the number of supplementary figures to those that are essential.Reviewer #3 (Recommendations for the authors):Recommendations to be addressed:– The whole paragraph 125–137 is very confusing in the wording between knock–out mice and conditional knock–out mice. The paragraph should clearly state that:i) First ABRs were recorded in knock–out mice…ii) We then recorded in ABRs in conditional knock–out mice…For instance, the use of "further" in line131 is misleading because it gives the impression we are dealing with the same mice whereas it is not the case. Similarly, what are Rest–homozygous mice? I suppose they are Rest knock–out mice?

We apologize for the confusing description. We have corrected the sentence based on the comment. We only used *Rest* conditional knockout mice, including heterozygotes (*Rest* +/cKO) and homozygotes (*Rest* cKO). (lines 128-141)

– While reading the manuscript, I thought the authors could better insist on the OHC deficits and their relationship to the hearing loss observed. The authors insist very much on the SGN excitability deficits. However, these deficits are already present in 1 month old mice with a similar severity to 3 month old mutant mice although ABR thresholds are still normal at 1 month of age. A likely explanation is rather the DPOAE deficit at three months of age that becomes very strong compared to one month. To better disentangle these phenomena, the authors should analyze the amplitude of wave–1and its timing in Rest cKO mice based on their already acquired data. If these parameters are normal at one month of age, it suggests that the most dramatic phenotype relies on OHCs. Alternatively, wave–1 amplitude could be altered (with normal auditory threshold). In that case, it would show that both the OHC and SGN deficits cumulatively contribute to the hearing impairment. In addition, Kcnq4 playing on the excitability of SGNs, one would expect the timing of wave–1 to be altered.Depending on these analysis, the authors should better discuss the relative contribution of SGN and OHCs to the auditory phenotype.

Thank you for your comments. We reanalyzed the amplitude and latency of wave I in 1-, 3- month-old WT and *Rest* cKO mice (Figure 1—figure supplement 2). In response to ABR pure-tone 16KHz stimuli at 70 dB SPL, the data revealed lower amplitude and increased wave I latency in both 1- and 3-month-old Rest cKO mice compared to the WT groups (Figure 1—figure supplement 2E, 2F), suggesting cumulative deficits in both OHC and SGN may contribute to hearing impairment.

– Line373: The statement "the exact site of action of systemic fasudil on hearing requires further elucidation" is not satisfactory. As mentioned earlier, by analyzing wave 1 amplitude and timing in their signal, the authors could specifically address whether fasudil affects HCs and SGNs. Expression of Kv7.4 in the dorsal cochlear nucleus is not expected to affect the ABR signal before Wave–II. Please better discuss this point after analyzing these parameters in the datasets.

Thank you for your suggestions. Our data show that fasudil-treated mice exhibit lower wave I and wave II amplitudes in response to ABR click stimulation at 70 dB SPL. In response to a 16 kHz pure tone stimulus at 70 dB SPL, fasudil-treated mice showed lower wave I and wave II amplitudes and increased delay times compared to controls (Figure 7—figure supplement 2). Based on our results we rewrote the discussion. (lines 377-381)

[Editors' note: further revisions were suggested prior to acceptance, as described below.]

The reviewers agree that the manuscript has been improved and will make a valuable contribution to the field. However, there are several remaining issues that need to be addressed, concerning both presentation and interpretation of the data, as outlined below. Please address these concerns, and make sure that any figures used in the rebuttal have also been added to the manuscript, as requested by Reviewer 1. Reviewer 2 also suggests that you may wish to consider moving some of the supplementary material into the main figures. Reviewer 3 also lists a number of typos to be corrected.Reviewer #1 (Recommendations for the authors):The authors of this manuscript conclude that genetic inactivation of Rest in hair cells (HCs) and spiral ganglion neurons (SGNs) cause progressive hearing loss by upregulating the expression of Kv7.4 in the two cell types ("Repressor element 1–silencing transcription factor deficiency yields profound hearing loss through Kv7.4 channel upsurge in auditory neurons and hair cells"). This conclusion is not fully supported by the presented data because the utilized Cre–expressing transgene and some of the key activators/inhibitors in the study are not sufficiently selective to draw a definitive conclusion: (1) The Atoh1–Cre transgene, which was used for the generation of Rest cKO mice in the study, is expressed not only in HCs and SGNs but also in many non–HCs and non–SGNs in the cochlea. This transgene is an excellent tool for many purposes but it is not ideal to determine the effects of SGN/HC specific deletion of Rest on hearing because Rest is expressed in non–HCs and non–SGNs in the hearing organ. (2) Fasudil, which was used for the activation of Kv7.4 in the study, is not a Kv7.4 specific drug; it also binds to (and inhibits) Rho–associated kinases. Thus, the hearing loss of fasudil–injected mice may not be caused by the activation of Kv7.4 but by the inhibition of Rho–associated kinases. (3) XE991 is not a specific inhibitor of Kv7.4, it also inhibits other Kv7 channels. Therefore, it is unclear whether the 10–15 dB improvement in the hearing threshold of XE991–treated Rest cKO mice is caused by the inhibition of Kv7.4 or other Kv7 channels. In summary, the authors' overall conclusion is not entirely convincing because much of their data can be interpreted differently than specified in the manuscript.

We appreciate the reviewer's concerns and agree with most of the above arguments. In the revised manuscript, we have acknowledged these limitations of the approaches taken. Yet, we think that our conclusion that REST deficiency impairs hearing via Kv7.4 upregulation is still the most likely explanation for our results.

1) As mentioned by the reviewer, atoh1 is not only expressed in SGNs and HCs but also in some supporting cells of the organ of Corti. Therefore, to determine whether supporting cells are involved in the hearing loss caused by *Rest* deletion, we examined the morphology of cochlear cells in 1- and 3-month-old WT and Rest cKO mice by HE staining. We did not find morphological abnormalities in the supporting cells (Data not shown in the manuscript). Although these results cannot completely rule out the possibility that deletion of *Rest* in support cells may also be involved in deafness, our current data are consistent with the hypothesis that SGNs and HCs are the main cell types contributing to hearing loss in *Rest* cKO mice.

2) Fasudil, a compound with Kv7.4 channel activator activity, also has an inhibitory effect on Rho-associated kinases. Recently, fasudil was found to prevent neomycin-induced hair cell damage by inhibiting the Rho/Rock signaling pathway, suggesting a cell protective effect as a Rock inhibitor (PMID: 31780893). Moreover, Seiji Kakehata's group did show that ROCK inhibition attenuated cochlear synaptic damage and exerted cytoprotective effects (PMID: 34217338, PMID: 33328888). In our study, injection of fasudil induced hearing loss in mice with increased Kv7 current in SGNs and HCs. Therefore, we reasoned that it might be caused by activation of the Kv7.4 channel rather than inhibition of Rock. Yet, the effect of the Rho pathway cannot be ruled out, which is now acknowledged.

3) In the cochlea, Kv7.1 is expressed in the marginal cells of the stria vascularis, Kv7.2 and Kv7.3 channels are present in the SGNs, and Kv7.4, one of the critical ion channels that regulate auditory signal, is mainly expressed in the SGNs and HCs. Since marginal cells in the stria vascularis have not been found to express Atoh1 (PMID: 20533400), specific deletion of *Rest* should not affect Kv7.1. Moreover, our results show that Kv7.2 and Kv7.3 in SGNs were not upregulated in *Rest* cKO mice. Therefore, we reasoned that XE991 improved hearing in *Rest* cKO mice, likely through inhibition of Kv7.4 channels.

In the authors' rebuttal letter (and in the manuscript), one unvalidated data interpretation is used to support another unvalidated interpretation; e.g., in the 3rd answer to reviewer–1, the authors view upregulation of Kv7.4 as a proven cause (as opposed to a 'possible cause') of hearing impairment in Rest cKO mice and use this to state that the hearing loss–relevant deletion of Rest must have been in HCs and/or SGNs because other cochlear cells do not express Kv7.4. This kind of reasoning goes awry easily.

We apologize for the poor choice of words to answer this question in the previous review comments. As mentioned by the reviewer, Atoh1 is widely expressed in cochlear; deletion of *Rest* is not limited to HC and SGN but also includes some supporting cells. Although our data suggest that Kv7.4 upregulation in HCs and SGNs is involved in hearing loss in *Rest* cKO mice, other possibilities exist. We modified the language throughout to acknowledge possible other explanations for the observed effects (we apologize for the apparent circular thinking).

The authors show the ABR thresholds (click and pure tone) of 3–4 months old Rest Het mice in the rebuttal letter but not in the actual manuscript. It would be more useful to the readers if these figure panels were added to the manuscript.

Thank you for your suggestion. We have added this data to figure1—figure supplement 1B, C.

Reviewer #2 (Recommendations for the authors):1. One of the most interesting aspects of this data set was the patch–clamping data showing enhanced expression of Kv7 currents in Rest cKO mice and the accompanying changes in ABR thresholds at 1M of age, before the onset of obvious cell deterioration. This combined with the experiments showing partial recovery of thresholds using the Kv7 blocker xe991 was the strongest indication that modulation of this particular ion channel was shaping hearing sensitivity, even in the absence of cell death. I had urged the authors to examine the stimulus level–dependent changes in ABR amplitudes and latencies to more fully characterize the hearing status under these different manipulations, hoping that this would help disentangle the impact of cell deterioration versus enhancement of Kv7 currents on the hearing loss phenotype. The authors have partially satisfied that request by providing more ABR data in supplements. However, by providing this analysis at a fixed level and with little further discussion, the authors missed the point…which was to test if there were differences in how the amplitude and latencies grow with stimulus intensity which might have indicated interesting differences in ABR morphology that could be attributed to the increase/decrease in phasic firing in Rest KO or XE199 manipulations. As currently presented in the text, the new ABR data does little to interpret the finding further.

We are very sorry for misinterpreting the previous review comments regarding further analysis of the ABR data. We have re-examined the stimulus level-dependent changes in ABR amplitudes and latencies in *Rest* cKO, XE991, and fasudil-treated mice. Our results showed that compared to WT mice, 1-month-old *Rest* cKO mice displayed stimulus level-dependent decreases in amplitudes and increases in latencies of ABR waves I and II, which was more pronounced in 3-month-old mice, suggesting a gradual decline in the hearing status of the *Rest* cKO mice (figure 1-supplementary figure 2). Fasudil treatment mice displayed shorter amplitudes and longer latencies of waves I and II, suggesting that fasudil might cause auditory nerve processing dysfunction (figure 7-supplementary figure 1). Moreover, increased amplitudes and decreased latencies of ABR waves I and II were shown in *Rest* cKO mice treated with 0.25 mg/kg XE991, indicating an improvement in their hearing (figure 8-supplementary figure 1).

2. Is elevation in Kv7 currents neuroprotective or does this lead to cellular deterioration? Lines 297–306 in the discussion left me with the impression that the authors are arguing that overexpression of Kv7 currents can cause a similar degeneration in cochlear cells as previously shown due to loss of function mutations in Kv7.4. This is perhaps not what the authors intended, because they agreed in their response to my original review that knocking out REST may cause cellular deterioration from multiple pathways and that the fasudil manipulation did not show cellular deterioration. Thus, these studies cannot determine if the cellular degeneration noted in the 3–4 month Rest KO mouse is due to overexpression of Kv7…so the question of whether Kv7 may be neuroprotective cannot be addressed here. I'd recommend a careful revision of this paragraph to avoid conflating elevation in threshold that they may attribute to the enhancement of Kv7 from that which may result from overall cellular damage.

We thank the reviewer for this suggestion; we have now revised this discussion paragraph. We proposed that upregulation of Kv7.4 channels is an important cause of hearing loss in 1-month-old *Rest* cKO mice in the absence of degeneration of SGN and HC. In contrast, 3-month-old *Rest* cKO mice showed significant SGN and HC deficits, and whether the upregulation of Kv7.4 channels caused them requires further experimental evidence in the future (lines 328-340).

3. The reduced recovery of thresholds for the higher dose of Xe991 is counterintuitive. Why doesn't the higher concentration of XE991 restore thresholds and only at lower frequencies as did the lower concentration? Rather than addressing what seems counterintuitive to me, the authors assert that there is an improvement in threshold and gloss over what seems to be only a partial rescue at the higher dose. Some sort of discussion of this is needed…for example is there some balance between synchrony and excitability that may produce this result? It will be important to do this since the rescue with the blocker is one of the most convincing pieces of data in this study.

We thank the reviewer for pointing this out. Yes, our results showed that low concentrations (0.25mg/kg) of XE991 significantly rescued hearing in *Rest* cKO mice, while higher concentrations (0.5mg/kg) of XE991 can only partially restore hearing in these mice. To discuss this apparent inconsistency, we have added the following Discussion. "XE991, a K_v_7 channel blocker, was used to demonstrate that upregulation of K_v_7.4 channels caused by REST defects is involved in deafness in mice. Our results indicated that at low concentrations (0.25 mg/kg), XE991 significantly rescued the hearing of *Rest* cKO mice, while at high concentrations (0.5 mg/kg) XE991 had a more modest effect on restoring the hearing of these mice. As a certain number of Kv7.4 channels is necessary to regulate cell excitability and thus maintain hearing in mice, their up or downregulation beyond the basal range of expression would impact hearing. Therefore, we hypothesize that 0.25 mg/kg XE991 could appropriately inhibit the overexpressed K_v_7.4 channels in *Rest* cKO mice, restoring them to a "safe" range.

In contrast, 0.5 mg/kg XE991 may over-inhibit K_v_7.4 channels beyond the range in which K_v_7.4 channels perform their normal function and fail to improve hearing in mice completely. Other on- or off-target side effects of higher doses of XE991 also cannot be excluded." (lines 418-430)

4. Problem with the summary figure shown in figure 9. The schematic shows an OHC providing direct input to Type I auditory neurons. Since the input from OHCs to most of the auditory nerve is indirectly coupled through fluid mechanics, this figure could be misconstrued as meaning otherwise. Please revise to avoid such confusion. Alternatively, remove the figure.

We agree with the issue pointed out by the reviewer that the Type I SGN did not form a direct link/synapse to the OHC and that our figure was misleading. Thus, we removed figure 9.

5. I still found several supplementary figures to be disruptive to the presentation of the material. Lines 178:186; Figure 3 supplement and Figure 5 supplement 1. I don't see the rationale for discussing the P14 data separately from the 1 Month and 3 Months time points. It seems that it would be natural to integrate all three time points into the same figure. Same comment for Figure 5, why not fold in the P14 data in rather than show a nearly identical representation of that one–time point as a supplement?

We thank the reviewer for the comments. We have integrated the data for P14, 1-, and 3-month-old mice in the main Figure 3 and Figure 5, respectively.